# DynamicVL: Benchmarking Multimodal Large Language Models for Dynamic City Understanding

**Weihao Xuan**[1,2*]  **Junjue Wang**[1*]  **Heli Qi**[2,3]  **Zihang Chen**[4]
**Zhuo Zheng**[5]  **Yanfei Zhong**[4]  **Junshi Xia**[2]  **Naoto Yokoya**[1,2†]
[1] The University of Tokyo  [2] RIKEN AIP  [3] Waseda University
[4] Wuhan University  [5] Stanford University

## Abstract

Multimodal large language models (MLLMs) have demonstrated remarkable capabilities in visual understanding, but their application to long-term Earth observation analysis remains limited, primarily focusing on single-temporal or bi-temporal imagery. To address this gap, we introduce **DVL-Suite**, a comprehensive framework for analyzing long-term urban dynamics through remote sensing imagery. Our suite comprises 14,871 high-resolution (1.0m) multi-temporal images spanning 42 major cities in the U.S. from 2005 to 2023, organized into two components: **DVL-Bench** and **DVL-Instruct**. The *DVL-Bench* includes six urban understanding tasks, from fundamental change detection (*pixel-level*) to quantitative analyses (*regional-level*) and comprehensive urban narratives (*scene-level*), capturing diverse urban dynamics including expansion/transformation patterns, disaster assessment, and environmental challenges. We evaluate 18 state-of-the-art MLLMs and reveal their limitations in long-term temporal understanding and quantitative analysis. These challenges motivate the creation of *DVL-Instruct*, a specialized instruction-tuning dataset designed to enhance models' capabilities in multi-temporal Earth observation. Building upon this dataset, we develop **DVLChat**, a baseline model capable of both image-level question-answering and pixel-level segmentation, facilitating a comprehensive understanding of city dynamics through language interactions. Project: `https://github.com/weihao1115/dynamicvl`.

## 1 Introduction

Sustainable city, as a key goal in "The 2030 Agenda for Sustainable Development"[3], has proposed new requirements for urban resilience, convenience, and comfort. Remote sensing technology enables us to monitor urban development over time by analyzing satellite imagery, allowing us to track large-scale changes in urban landscapes [8, 45]. However, research in this field has been largely limited to comparing images from only two time points [54, 42], primarily due to the scarcity of well-aligned vision-language datasets spanning longer time series. This limitation has constrained our ability to conduct a comprehensive, large-scale understanding of urban dynamics.

The recent emergence of MLLMs [20, 26, 40] represents a significant advancement in visual-language understanding. These models mark a shift from specialized, single-purpose systems to versatile frameworks capable of handling multiple tasks, including but not limited to visual grounding [27], image captioning [5], and visual question answering [51, 52]. While recent MLLM research has demonstrated promising results in multi-image [29, 15] and video understanding [43, 12], these

---

[*] Equal contribution.

[†] Corresponding author.

[3] https://sdgs.un.org/goals/goal11

39th Conference on Neural Information Processing Systems (NeurIPS 2025) Track on Datasets and Benchmarks.

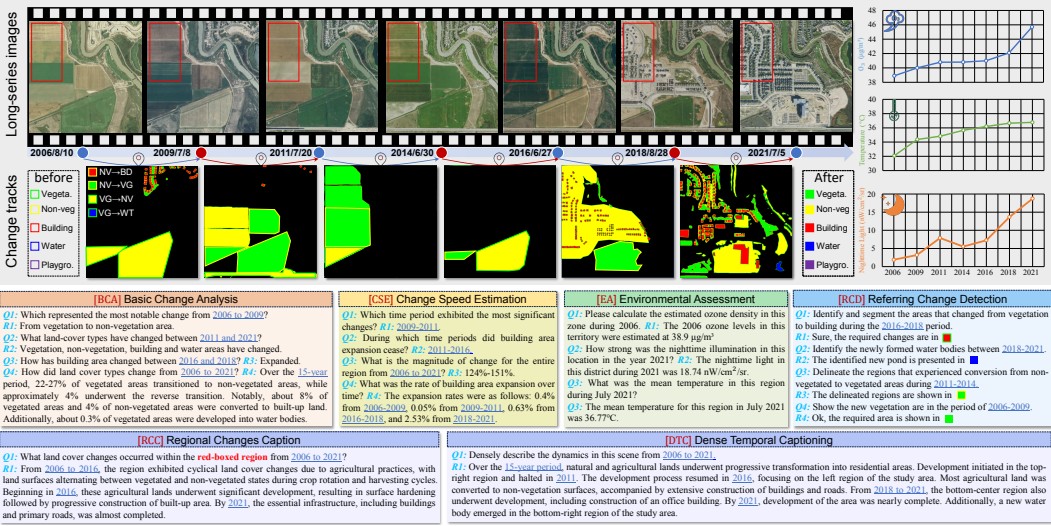

Figure 1: Diverse tasks in the **DVL-Bench**. Our framework encompasses multiple levels of temporal understanding: from pixel-precise change detection and quantification to regional evolution analysis and dense temporal captioning. This hierarchical task design enables systematic evaluation of MLLMs' capabilities in multi-temporal Earth observation understanding.

efforts primarily focus on in-the-wild daily imagery. In the context of remote sensing, existing research [13, 37] on multi-temporal analysis is typically limited to bi-temporal or short-sequence comparisons. Moreover, current multi-temporal MLLMs in remote sensing are primarily tested on high-level semantic understanding, lacking pixel-precise analysis capabilities crucial for quantitative change assessment. These limitations pose significant challenges for dynamic city understanding applications, which require both long-term understanding beyond bi-temporal inputs and precise quantitative analysis of environmental changes.

To address these challenges, we present **DVL-Suite**, a comprehensive framework for analyzing urban dynamics over time using remote sensing imagery. Our framework introduces **DVL-Bench**, a large-scale benchmark designed for rigorous evaluation of vision-language models in urban contexts. Building on recent advances in video understanding benchmarks [36, 27, 49], we develop a structured taxonomy that identifies six core capabilities essential for sustainable urban understanding: 1) [BCA] Basic Change Analysis: Focuses on identifying and comparing multi-temporal changes in land-use patterns. 2) [CSE] Change Speed Estimation: Tracks and quantifies temporal trends of key urban elements, such as building expansion rates and vegetation loss. 3) [EA] Environmental Assessment: Evaluates urban livability and economic indicators through visual analysis. 4) [RCD] Referring Change Detection: Tests models' capabilities in dense reasoning and precise spatial localization of changes. 5) [RCC] Regional Change Captioning: Generates detailed change descriptions for user-specified geographical areas. 6) [DTC] Dense Temporal Captioning: Generates comprehensive reports documenting long-term temporal changes, highlighting critical events across long time series.

To ensure comprehensive coverage, DVL-Bench encompasses diverse urban scenarios, such as urban expansion, housing crises, natural disasters, urban heat island effects, and green space development. Unlike traditional bi-temporal understanding, it enables systematic evaluation of long-term city dynamics, facilitating deeper insights into sustainable city development through multi-temporal Earth observation.

Through extensive experimentation on DVL-Bench, we discovered that state-of-the-art MLLMs, both commercial and open-source models, face significant challenges in long-term temporal visual understanding, primarily due to insufficient training data spanning extended time periods. To address these limitations, we introduce **DVL-Instruct**, a specialized instruction-tuning dataset designed for dynamic city understanding in remote sensing. Using this dataset, we develop **DVLChat**, a baseline model that enhances multi-temporal urban understanding and caption generation, while introducing referring change detection capabilities.

The key contributions of this paper are as follows:

1. We introduce DVL-Suite, comprising DVL-Bench and DVL-Instruct, with 14,871 high-resolution (1.0m) images spanning 42 U.S. cities, featuring an average of 6.73-6.94 temporal frames per scene from 2005 to 2023, enabling long-term urban dynamics analysis with

a coherent task taxonomy, multi-level analysis capabilities, and thematic focus on urban development patterns.

2. We evaluate 18 vision-language models, revealing critical limitations: the best-performing model, o4-mini, achieves only 34.1% accuracy on DVL-Bench's overall QA average, demonstrating significant deficiencies in complex temporal tasks and quantitative analysis.

3. Based on DVL-Instruct, we develop DVLChat, a baseline that surpasses its base Qwen2.5-VL 7B by significant improvements, enabling multi-temporal urban analysis and referring change detection from a single model.

## 2 Related Work

### 2.1 Large Multimodal Language Models

The rapid advancement of MLLMs has sparked significant interest in their applications to complex visual understanding tasks, including understanding temporal dynamics and multiple image inputs, which are central to multi-temporal remote sensing analysis. In generic vision-language models, early pioneering works such as Flamingo [1] demonstrated the capability to process interleaved sequences of visual and textual data, including video frames, through mechanisms like Perceiver Resampler. Subsequent developments have enhanced video understanding, with models like Video-LLaVA [24] unifying image and video representations before LLM projection, and LLaVA-OneVision [20] offering a unified framework for single-image, multi-image, and video tasks via the "Higher AnyRes" strategy. Qwen2-VL series [40, 2] introduced Multimodal Rotary Position Embedding (M-ROPE) aligned with absolute time for long video comprehension, and InternVL3 [55] employed Variable Visual Position Encoding (V2PE) for extended multimodal contexts including lengthy video sequences. Concurrently, the challenge of multi-image understanding has been addressed by models such as LLaVA-NeXT-Interleave [21], which utilizes a data-centric approach with the M4-Instruct dataset to handle diverse multi-image scenarios.

Despite these advances in temporal and multi-image processing, existing models still fall short in dynamic, long-term remote sensing analysis, particularly for precise quantitative assessment of urban changes. To address this gap, we introduce DVL-Suite and DVLChat to advance multi-temporal urban understanding.

Table 1: Comparison with existing multi-temporal remote sensing vision-language datasets.

| Dataset | Self-contained | Average Temp. | Text Pairs | MT Images | Image Size | [BCA] | [CSE] | [EA] | [RCD] | [RCC] | [DTC] |
|---|---|---|---|---|---|---|---|---|---|---|---|
| RSICap [28] | ✓ | 1 | 2585 | 0 | 512 | × | × | × | × | × | × |
| LHRS-Bot [30] | ✓ | 1 | 1.2M | 0 | 768 | × | × | × | × | × | × |
| VRSBench [23] | ✓ | 1 | 205k | 0 | 512 | × | × | × | × | × | × |
| GeoChatSet [17] | × | 1 | 318k | 0 | 504 | × | × | × | × | × | × |
| CDVQA [50] | ✓ | 2 | 122k | 122k | 512 | ✓ | × | × | × | × | × |
| LEVIR-MCI [25] | ✓ | 2 | 50.3k | 50.3k | 256 | × | × | × | × | × | × |
| OVG-360k [22] | × | 2 | 360k | 360k | 512 | ✓ | × | × | ✓ | × | × |
| ChangeChat [6] | × | 2 | 87k | 87k | 256 | ✓ | × | × | × | × | × |
| GeoLLaVA [7] | × | 2 | 100k | 100k | 336 | × | × | × | × | × | × |
| CC-Expert [41] | × | 2 | 135k | 135k | 384 | × | × | × | × | × | × |
| TEOChatlas [13] | × | 2.07 | 554k | 245k | 224 | ✓ | × | × | × | ✓ | × |
| EarthDial [37] | × | 1.01 | 11M | 64.6k | 448∼1024 | ✓ | × | × | × | ✓ | × |
| **DVL-Bench** | ✓ | **6.94** | 8,682 | 3,469 | **1024** | ✓ | ✓ | ✓ | ✓ | ✓ | ✓ |
| **DVL-Instruct** | ✓ | **6.73** | 63,771 | 11,402 | **1024** | ✓ | ✓ | ✓ | ✓ | ✓ | ✓ |

### 2.2 Multimodal Benchmarks in Remote Sensing

Remote Sensing (RS) domain has witnessed the emergence of numerous specialized multimodal datasets [44]. LHRS-Bot [30], VRSBench [23], and GeoChatSet [17] pioneered single-temporal instruction datasets for classification, detection, and visual question answering (VQA). Subsequently, CDVQA [50] introduced change-aware VQA, while LEVIR-MCI [25] integrated pixel-level masks. GeoLLaVA [7] and CC-Expert [41] enhanced interactive bi-temporal captioning, and OVG-360k [22] provided fine-grained spatial semantic supervision. Although surpassing single-temporal analyses, these efforts remain limited to bi-temporal image pairs. Recently, TEOChatlas [13] curates temporal instruction-following tasks such as those derived from xBD [10] and fMoW [4]. DisasterM3 [39] provides a multi-hazard, multi-sensor, and multi-task remote sensing vision-language benchmark

with 26,988 bi-temporal satellite images and diverse disaster assessment tasks. However, existing datasets predominantly focus on bi-temporal understanding and lack comprehensive evaluations of models' capabilities in processing extended temporal sequences and performing long-term spatiotemporal reasoning. To enable MLLM to excel in understanding long-term remote sensing images, we introduce DVL-Bench, a large-scale vision-language benchmark for remote sensing analysis within long time series that offers three key advantages: **1) Coherent task taxonomy.** Unlike composite datasets assembled from heterogeneous sources, DVL-Bench introduces a systematically designed task taxonomy built upon newly collected data with consistent annotation standards. **2) Diverse temporal tasks.** DVL-Bench includes multiple analysis granularities, progressing from fine-grained pixel-level change detection and region-based dynamic captioning to holistic temporal reasoning and comprehensive environmental assessment, thereby facilitating systematic city dynamics understanding. **3) Practical and thematically focused.** In contrast to existing datasets addressing wide-ranging geospatial tasks, DVL-Bench specifically targets the analysis and representation of long-term urban development dynamics. In addition, the developed DVLChat can be directly integrated with the NAIP platform, serving as an AI assistant for urban understanding applications.

## 3 DVL-Suite Curation Pipeline

To ensure data diversity and quality, we built DVL-Suite using high-resolution (1.0m GSD) remote sensing imagery from the National Agriculture Imagery Program (NAIP), covering 42 major U.S. cities. The imagery was first geo-referenced and processed into 14,871 patches of $1024 \times 1024$ pixels, comprising 2,193 multi-temporal scenes. For compatibility with generic MLLMs, we utilized three optical bands. By collecting environmental datasets from diverse Earth observation platforms (sources are detailed in Appendix § D), all data were spatially resampled to match the resolution of collected remote sensing imagery, ensuring one-to-one correspondence between images and environmental indicators. Based on the multi-source Earth observation data, we hired several experts and a well-trained annotation team to label and examine the DVL-Suite.

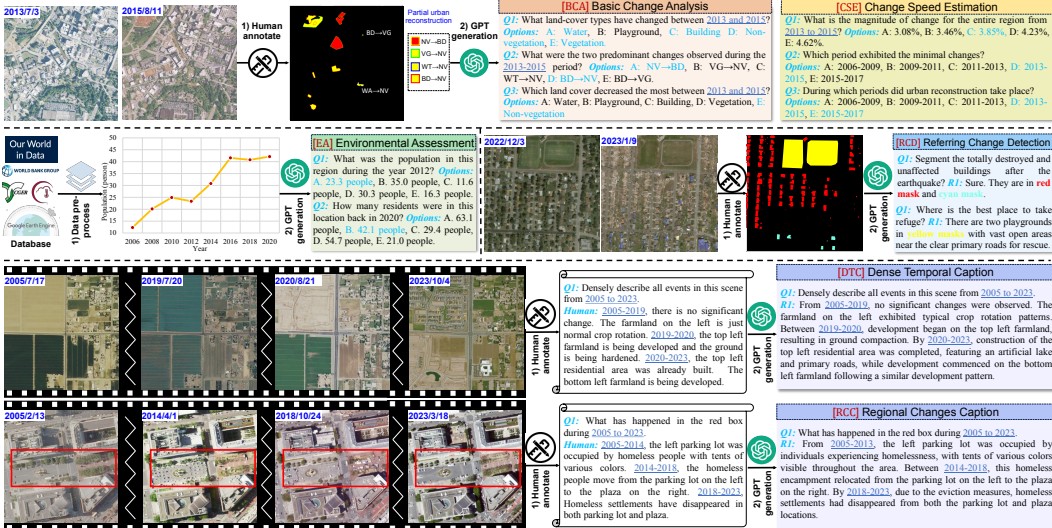

Figure 2: The annotation pipeline of the proposed DVL-Suite. Four common urban dynamics are depicted from top to bottom: partial urban reconstruction, natural disasters, farmland conversion, and homeless encampments. In our semi-auto pipeline, urban experts perform the basic annotations, while GPT4.1 integrates this information to generate the paired instructions.

Figure 2 illustrates our multi-stage annotation pipeline for the DVL-Bench dataset. The annotation process includes several specialized tasks: For bi-temporal analysis tasks ([BCA] and [CSE]), annotators first segmented semantic change areas between adjacent temporal images. These changes were categorized across five primary land-cover types: vegetation, non-vegetated surfaces, water, buildings, and playgrounds. This categorization, adapted from SECOND [46], yielded 20 distinct change event categories. GPT4.1 then generated diverse task-specific instructions using these segmentation masks and categories. For [BCA] questions (e.g., "What land-cover type changed most between 2015 and 2017?"), the system calculated correct answers from the masks and generated

four incorrect options from other land-cover types. For [CSE] tasks, change speeds were computed from the masks, with alternative options varying by ±20% and ±40%. The [EA] task followed a similar pipeline, but utilized the multi-source environmental indicators as reference data. For [RCD] tasks, domain experts designed event-specific prompts, followed by manual mask annotation and GPT4.1-based language enhancement of prompts and answers. For temporal narrative tasks ([DTC] and [RCC]), annotators first identified keyframes containing significant changes to segment the temporal sequence, then crafted period-specific captions. GPT4.1 refined these draft captions to enhance detail and coherence. While both tasks follow identical procedures, [RCC] focuses on user-specified local areas rather than global changes.

After the first round of labeling, we conducted a rigorous quality control process: self-examination, cross-examination to correct errors (including false labels, missing details, and inaccurate captions), and supervisor review of 1,000 randomly sampled annotations. Any unqualified annotations were returned for refinement. DVL-Instruct follows the same data curation pipeline, but differs in that it exclusively pairs instructions with ground truths rather than providing multiple-choice options.

## 4 DVL-Bench Designs

DVL-Bench consists of 3,469 multi-temporal Earth observation images, each accompanied by human-verified annotations. The annotation suite includes 1,391 referring segmentation instructions for precise change localization, 5,854 question-answer pairs for detailed temporal analysis, and 1,437 comprehensive captions documenting urban dynamics. This section outlines the task taxonomy and examines the fundamental challenges in interpreting long-term remote sensing sequences.

**Instruction distributions with different tasks.** Figure 3 presents the distribution of instructions across task categories. The [BCA], [CSE], [EA], and [RCD] tasks exhibit relatively balanced sample distributions, collectively accounting for 89.9% of the benchmark. In contrast to these tasks, which primarily focus on part of frames within each scene, the [DTC] and [RCC] tasks require comprehensive analysis of complete temporal sequences, thus representing smaller proportions of the overall distribution. With substantial sample sizes across all categories, DVL-Bench enables systematic evaluation of MLLMs in both dynamic understanding and open-ended generation performances.

**Geospatial and temporal distributions.** Figure 4 shows the spatial distribution across 42 U.S. cities and the temporal length per scene. The samples are evenly distributed in 42 rapidly growing cities, ensuring comprehensive geographical coverage and minimizing regional bias. Unlike the existing multi-temporal dataset focusing on understanding bi-temporal changes, the DVL-Bench features long-term analysis with sequences ranging from five to ten frames per urban scene. This extended temporal scope poses new challenges for modeling discrete temporal transitions in urban environments.

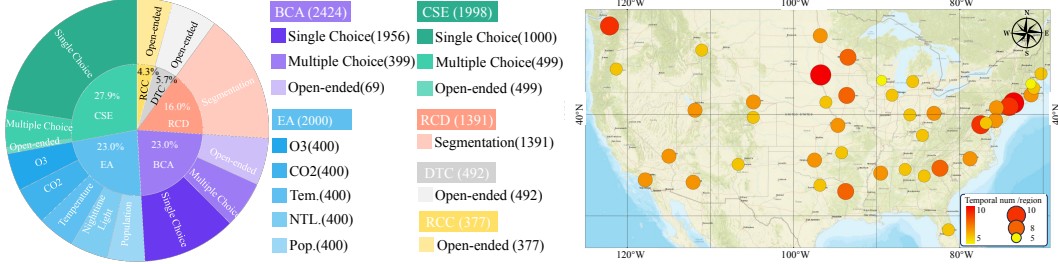

Figure 3: Task taxonomy and sample distribution in DVL-Bench. The multi-level task evaluates MLLM comprehensively.

Figure 4: Data distributions across the 42 rapidly growing cities and the temporal number floats from five to ten.

**Basic change analysis (BCA).** The Sankey diagram in Figure 5 quantifies land cover transition patterns between initial and final states. The visualization reveals a dominant trend of vegetation conversion to developed areas, reflecting rapid urbanization processes. In contrast, only a modest area of building, approximately $2.34km^2$ underwent demolition, primarily transitioning to vegetation and non-vegetation surfaces. These complex dynamics, involving multi-directional transitions among five distinct land cover types and requiring precise quantification across various spatial scales, present significant challenges for MLLMs in terms of both semantic understanding and numerical reasoning.

**Change speed estimation (CSE).** The temporal analysis in Figure 6 tracks building expansion rates across successive periods, providing insights into U.S. urban development trajectories over the past two decades. Development velocity exhibits a distinct non-linear pattern: accelerating from 2010, reaching peak urbanization rates around 2017, and showing significant deceleration after 2018. This characteristic growth curve, with its pronounced acceleration and subsequent slowdown, represents a typical urban development cycle. Such complex temporal dynamics require MLLMs to maintain precise numerical sensitivity while modeling long-term spatiotemporal variations.

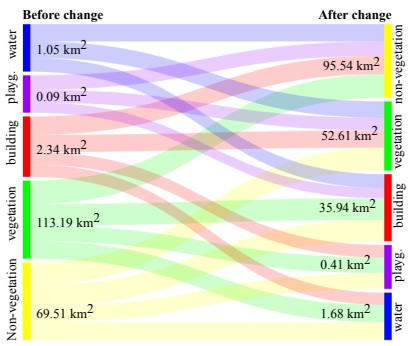

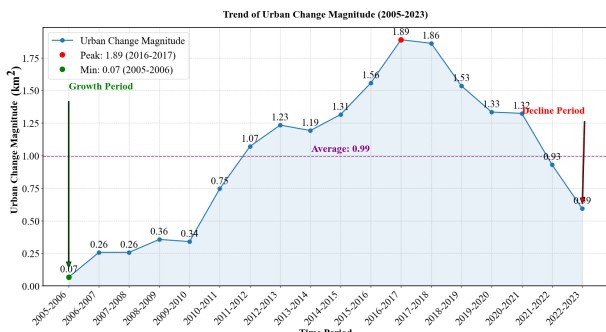

Figure 5: The basic change flow in DVL-Bench.

Figure 6: Trend in change magnitude per period, showing non-linear development speed across the U.S.

**Referring change detection (RCD).** Analysis of change scales in Figure 7 reveals distinct patterns across the five primary land-cover types. Changes in vegetation and non-vegetation areas show high variability, with a clear asymmetric distribution favoring smaller spatial extents. Buildings, water bodies, and recreational areas predominantly undergo small-scale changes, consistent with their limited spatial footprint in urban landscapes. These diverse change scales across land-cover types test MLLMs' ability to detect and segment temporal changes at varying spatial scales.

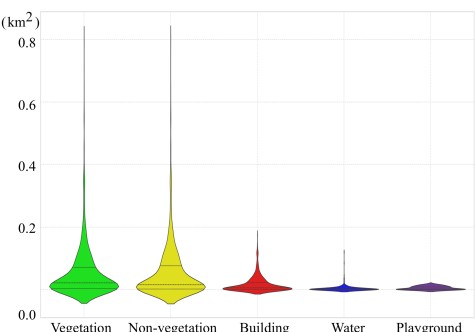

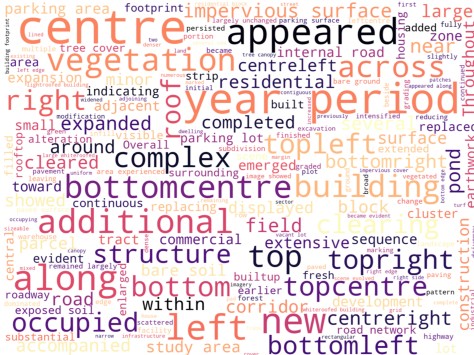

Figure 7: The change scale distributions in referring change detection.

Figure 8: Word cloud in the dense temporal and regional change captioning tasks.

**Dense temporal and region-level change caption (DTC and RCC).** The word cloud visualization in Figure 8 illustrates the distribution of the 300 most frequent terms in designed long-form captions. While existing geospatial vision-language datasets primarily focus on spatial descriptors (topright, left, etc.), DVL-Bench distinctively features rich temporal references (year, period, etc.) and transition vocabulary (replaced, added, completed, etc.), enabling comprehensive documentation of urban dynamics.

## 5 Benchmark Experiments

In this section, we present the experimental setup, introduce DVLChat as our baseline model for dynamic urban understanding, and comprehensively evaluate 18 state-of-the-art MLLMs on DVL-Bench.

## 5.1 Implementation Details

**Benchmark methods.** We evaluate 18 widely-adopted MLLMs across different capabilities: (1) open-source MLLMs, including domain-specific models like TEOChat [13] and EarthDial [37], state-of-the-art MLLMs with multi-image and video perception abilities (Video-LLaVA [24], LLaVA-OneVision [20], InternVL3 [55], and Qwen2.5-VL [2]), and referring segmentation models (LISA [19], PSALM [53]); (2) commercial MLLMs, including o4-mini [34], GPT4.1 [33], GPT4o [31], and Gemini 2.5 Flash [9]. Unless otherwise specified, all experiments using open-source models were conducted on 8 H100 GPUs. For question-answering tasks, we utilized each model's native multi-image inference capability. For referring change detection tasks, LISA and PSALM were evaluated using a different approach, concatenating two temporal images into a single composite image as input. The detailed training and testing methods can be found in the Appendix § A.

**Evaluation metrics.** Our evaluation framework employs multiple metrics to assess model performance across different tasks comprehensively. For the evaluations presented in Table 2, we measure both basic change analysis and change speed estimation using two approaches. First, we calculate accuracy percentages for single and multiple-choice questions. Second, for open-ended generation tasks, we evaluate Basic Change Reports using three metrics: Land Cover Type Identification (LCT), Time Period Accuracy (TPA), and Change Quantification Accuracy (CQA). Similarly, Change Speed Reports are assessed using Change Rate Precision (CRP), Time Period Accuracy (TPA), and Change Pattern Accuracy (CPA). For the long-form captioning tasks shown in Table 3, Regional Change Captioning is evaluated using Temporal Coverage (TC), Spatial Accuracy (SA), Process Fidelity (PF), and Region Containment (RC), while Dense Temporal Captioning uses TC, SA, and PF. All captioning metrics are scored on a 0-5 scale, with higher scores indicating better performance. These scores are determined by GPT4.1-mini [32] through comparison with reference captions. The detailed evaluation prompts can be found in the Appendix § B.2.

**DVLChat design.** As dynamic urban understanding necessitates both semantic comprehension and fine-grained pixel-level understanding, we followed the main architecture of LISA [19] to develop DVLChat. However, the original LISA model was unable to perform pixel-level segmentation while maintaining high open-ended capabilities due to optimization conflicts. Furthermore, LISA, designed for single-image analysis, lacked the ability to analyze changes across multiple images, whereas the DVL-Instruct data enables these capabilities. Therefore, leveraging DVL-Instruct, we develop DVLChat as a baseline model for multi-temporal urban understanding tasks. As shown in Figure 9, DVLChat employs a task-specific routing mechanism through dedicated prefix tokens from users. The system routes user queries to specialized modules based on their prefixes:

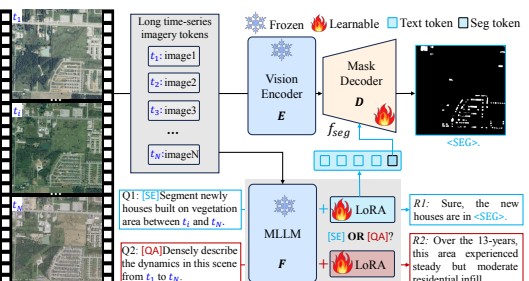

Figure 9: Detailed illustration of DVLChat. We separate question-answering and referring change detection by implementing two distinct LoRA [11] modules, enabling the model to possess independent VQA and segmentation capabilities while preventing interference between their respective data streams in the training.

inputs with [QA] activate the VQA LoRA module for generating textual responses, while those with [SE] engage the change detection LoRA module. DVLChat addresses multi-temporal analysis by interleaving image features from multiple temporal images before decoding. For referring change detection tasks, the system processes this interleaved representation and decodes the <SEG> token embedding using SAM's [16] frozen vision backbone and unfrozen decoder to generate precise segmentation masks. DVLChat effectively isolates question-answering and change detection functionalities, preventing task interference in the original LISA algorithm while maintaining model efficiency. While our implementation uses Qwen2.5-VL as the MLLM, the architecture is MLLM-agnostic and can accommodate other multimodal language models.

## 5.2 Benchmark Results

**Overall benchmark performance.** As shown in Table 2, quantitative results reveal significant challenges in understanding urban dynamics through remote sensing imagery, with all current models

Table 2: Quantitative evaluation results of various vision-language models on basic change analysis (BCA), change speed estimation (CSE), and environmental assessment (EA) tasks. Performance is measured by accuracy percentages for single/multiple-choice questions and a scale of 0-5 for captioning tasks.

| Method | AVG | BCA-QA | | CSE-QA | | EA | BCA-Report | | | | CSE-Report | | | |
|---|---|---|---|---|---|---|---|---|---|---|---|---|---|---|
| | | Single | Multi | Single | Multi | | AVG | LCT | TPA | CQA | AVG | CRP | TPA | CPA |
| **Commercial models** | | | | | | | | | | | | | | |
| o4-mini [34] | 34.1 | 62.8 | 36.1 | 33.8 | 12.4 | 25.3 | 3.16 | 2.85 | 4.70 | 1.93 | 2.34 | 0.97 | 3.71 | 2.33 |
| GPT4.1 [33] | 32.5 | 66.1 | 39.7 | 31.3 | 5.4 | 20.2 | 3.02 | 2.69 | 4.67 | 1.72 | 2.23 | 0.78 | 3.84 | 2.05 |
| GPT4o [31] | 29.7 | 63.3 | 19.3 | 32.3 | 7.3 | 26.2 | 2.96 | 2.55 | 4.66 | 1.66 | 2.21 | 0.73 | 3.46 | 2.43 |
| Gemini 2.5 Flash [9] | 24.4 | 46.3 | 15.8 | 21.0 | 12.1 | 26.8 | 2.90 | 2.40 | 4.69 | 1.62 | 2.19 | 0.70 | 3.78 | 2.09 |
| **Open-source models** | | | | | | | | | | | | | | |
| TEOChat [13] | 17.2 | 35.1 | 8.7 | 17.0 | 10.8 | 14.6 | 0.64 | 1.61 | 0.22 | 0.09 | 1.22 | 0.85 | 1.46 | 1.33 |
| EarthDial [37] | 30.3 | 62.2 | 20.3 | 30.9 | 12.2 | 25.9 | 1.10 | 2.57 | 0.01 | 0.72 | 1.03 | 0.85 | 0.74 | 1.50 |
| Video-LLaVA [24] | 17.7 | 34.8 | 10.4 | 17.7 | 5.4 | 20.2 | 2.01 | 1.58 | 3.14 | 1.33 | 1.63 | 0.86 | 2.48 | 1.54 |
| LLaVA-OneVision 7B [20] | 19.3 | 41.7 | 2.8 | 21.5 | 4.8 | 25.9 | 2.30 | 2.29 | 3.20 | 1.42 | 1.72 | 0.95 | 2.44 | 1.78 |
| LLaVA-OneVision 72B [20] | 25.0 | 59.9 | 6.5 | 25.9 | 6.2 | 26.5 | 3.01 | 2.70 | 4.52 | 1.83 | 2.05 | 0.93 | 3.39 | 1.83 |
| InternVL3 8B [55] | 23.9 | 55.2 | 11.5 | 22.0 | 7.6 | 23.1 | 2.99 | 2.49 | 4.68 | 1.78 | 2.15 | 0.95 | 3.31 | 2.20 |
| InternVL3 14B [55] | 27.2 | 63.2 | 15.3 | 28.8 | 4.0 | 24.9 | 3.02 | 2.61 | 4.72 | 1.74 | 2.36 | 0.97 | 3.65 | 2.48 |
| InternVL3 78B [55] | 27.1 | 60.5 | 14.5 | 28.3 | 8.6 | 23.6 | 3.04 | 2.74 | 4.59 | 1.80 | 2.25 | 0.82 | 3.87 | 2.06 |
| Qwen2.5-VL 3B [2] | 24.7 | 56.9 | 6.0 | 26.1 | 9.2 | 25.1 | 2.99 | 2.72 | 4.58 | 1.65 | 1.72 | 0.57 | 3.42 | 1.18 |
| Qwen2.5-VL 7B [2] | 23.3 | 54.6 | 4.8 | 28.5 | 13.6 | 15.0 | 2.94 | 2.49 | 4.70 | 1.62 | 1.73 | 0.25 | 3.90 | 1.05 |
| Qwen2.5-VL 32B [2] | 31.4 | 62.0 | 33.3 | 36.9 | 3.2 | 21.6 | 3.04 | 2.65 | 4.65 | 1.81 | 2.60 | 1.21 | 3.89 | 2.71 |
| Qwen2.5-VL 72B [2] | 29.7 | 65.4 | 24.3 | 34.6 | 4.0 | 20.2 | 2.99 | 2.61 | 4.64 | 1.71 | 2.27 | 0.72 | 3.76 | 2.33 |
| **Ours** | | | | | | | | | | | | | | |
| DVLChat 7B | 33.3 | 64.9 | 21.3 | 31.3 | 18.6 | 30.6 | 3.47 | 3.41 | 4.72 | 2.28 | 2.51 | 1.48 | 3.41 | 2.65 |

demonstrating limited capabilities. The highest averaged accuracy of multiple-choice questions reaches merely 34.1% with o4-mini, while Qwen2.5-VL 32B and GPT4.1 achieve 31.4% and 32.5% respectively. Notably, TEOChat, despite being specifically designed for multi-temporal remote sensing vision-language tasks, achieves only 17.2% overall accuracy, struggling significantly with the benchmark's larger and city-level understanding compared to its native $256 \times 256$ input size. In contrast, our DVLChat 7B, leveraging the proposed DVL-Instruct dataset, demonstrates competitive performance at 33.3% while maintaining referring change detection capabilities.

**Task-specific challenges.** Diverse tasks evaluate MLLMs' performances from different aspects. While [BCA] with single-choice questions shows promising results, where Qwen2.5-VL 72B achieves 65.4% accuracy, performance degrades substantially in multi-choice settings, where even the leading model GPT4.1 only reaches 39.7%. The challenges become more pronounced in detailed analytical tasks. [BCA] report metrics expose fundamental limitations in both land cover type identification (LCT) and change quantification (CQA), with LCT scores maxing at 2.85 and CQA not exceeding 1.93 across existing models. Notably, by leveraging DVL-Instruct's comprehensive training data, DVLChat achieves a significant breakthrough in LCT with a score of 3.41, enhancing the recognition of changed land-cover types. [CSE] results reveal a critical limitation of current MLLMs in pixel-level change perception, with multi-choice accuracy peaking at merely 13.6% and Change Rate Precision (CRP) consistently below 1.21, indicating models' inability to capture and quantify fine-grained temporal variations. [EA] results are similarly concerning: except for our DVLChat 7B, other models achieve accuracies ranging from 14.6% to 26.8%, with many performing at or even below random chance (20% for 5-option questions).

**Captioning capabilities.** Table 3 reveals a substantial gap between commercial and open-source models in detailed captioning tasks. For the regional change captioning task, commercial models demonstrate superior performance with o4-mini achieving an average score of 4.58, while the best open-source model, InternVL3 14B, reaches only 3.96. Our DVLChat, incorporating DVL-Instruct, demonstrates strong performance with an average score of 3.98, approaching the performance of commercial models with 7B parameters. The disparity becomes even more pronounced in dense temporal captioning, where commercial models maintain strong performance with o4-mini reaching an average score of 4.14, while open-source alternatives struggle considerably with scores below 3.40. Notably, TEOChat achieves only 1.45, revealing severe limitations in handling complex temporal dynamics beyond bi-temporal comparisons.

**Scaling parameters.** Analysis of model size scaling reveals inconsistent improvements within different model families, which is distinguished from most generic computer vision tasks [52, 12]. Particularly in basic change analysis and change speed estimation tasks, the Qwen2.5-VL series shows notable improvements as model size increases to 32B, reaching 31.4% average accuracy, but performance declines to 29.7% with the 72B counterpart. Similarly, while LLaVA-OneVision

Table 3: Performance comparison of different models on regional change captioning (RCC) and dense temporal captioning (DTC) tasks, evaluated using Temporal Coverage (TC), Spatial Accuracy (SA), Process Fidelity (PF), and Region Containment (RC) metrics on a 0-5 scale.

| Method | RCC | | | | | DTC | | | |
|---|---|---|---|---|---|---|---|---|---|
| | AVG | TC | SA | PF | RC | AVG | TC | SA | PF |
| **Commercial models** | | | | | | | | | |
| o4-mini [34] | 4.58 | 4.79 | 4.21 | 4.35 | 4.97 | 4.14 | 4.64 | 4.04 | 3.73 |
| GPT4.1 [33] | 4.46 | 4.74 | 3.99 | 4.16 | 4.97 | 3.98 | 4.53 | 3.75 | 3.65 |
| GPT4o [31] | 4.32 | 4.66 | 3.78 | 3.89 | 4.96 | 3.87 | 4.45 | 3.65 | 3.49 |
| Gemini 2.5 Flash [9] | 4.34 | 4.66 | 3.84 | 3.87 | 4.99 | 3.61 | 4.15 | 3.41 | 3.28 |
| **Open-source models** | | | | | | | | | |
| TEOChat [13] | 1.66 | 1.02 | 0.45 | 0.29 | 4.87 | 1.45 | 1.65 | 1.14 | 1.57 |
| EarthDial [37] | 1.53 | 0.68 | 0.39 | 0.17 | 4.86 | 0.90 | 0.80 | 1.16 | 0.75 |
| Video-LLaVA [24] | 2.49 | 1.21 | 1.93 | 2.04 | 4.81 | 1.76 | 2.38 | 1.57 | 1.34 |
| LLaVA-OneVision 7B [20] | 3.07 | 3.12 | 2.37 | 2.17 | 4.63 | 2.17 | 2.36 | 2.09 | 2.08 |
| LLaVA-OneVision 72B [20] | 3.60 | 3.91 | 2.77 | 2.80 | 4.93 | 2.87 | 3.51 | 2.56 | 2.54 |
| InternVL3 8B [55] | 3.69 | 3.99 | 3.02 | 2.99 | 4.76 | 2.97 | 3.57 | 2.71 | 2.64 |
| InternVL3 14B [55] | 3.96 | 4.33 | 3.25 | 3.36 | 4.91 | 3.22 | 3.84 | 2.96 | 2.85 |
| InternVL3 78B [55] | 3.92 | 4.18 | 3.34 | 3.18 | 4.97 | 3.33 | 3.98 | 3.01 | 2.99 |
| Qwen2.5-VL 3B [2] | 2.76 | 2.77 | 1.82 | 1.52 | 4.92 | 2.38 | 2.38 | 2.66 | 2.11 |
| Qwen2.5-VL 7B [2] | 3.21 | 3.30 | 2.42 | 2.20 | 4.92 | 2.85 | 3.47 | 2.57 | 2.51 |
| Qwen2.5-VL 32B [2] | 3.90 | 4.28 | 3.18 | 3.23 | 4.92 | 2.91 | 3.39 | 2.77 | 2.57 |
| Qwen2.5-VL 72B [2] | 3.89 | 4.24 | 3.24 | 3.17 | 4.90 | 3.28 | 3.94 | 2.95 | 2.95 |
| **Ours** | | | | | | | | | |
| DVLChat 7B | 3.98 | 4.33 | 3.28 | 3.41 | 4.92 | 3.40 | 4.04 | 3.13 | 3.02 |

improves from 19.3% to 25.0% when scaling from 7B to 72B, InternVL3 peaks at 27.2% with its 14B variant before slightly declining to 27.1% with the 78B model. These non-monotonic scaling patterns in analytical tasks contrast sharply with the consistent improvements observed in captioning tasks, where larger models consistently achieve better performance in both regional and dense temporal captioning. This divergence in scaling behavior suggests that while model size benefits language generation and temporal narrative abilities, merely increasing parameters is insufficient for enhancing precise change detection and quantification capabilities. This is further evidenced by our DVLChat 7B outperforming larger models (up to 78B parameters) across multiple tasks when trained with domain-specific data. This highlights that incorporating strategies into domain-specific data is crucial for advancing model capabilities in understanding the analytical aspects of urban dynamics. We provide more analysis on scaling patterns by incorporating domain-specific data in Appendix § E.

**Referring change detection analysis.** We compare DVLChat with the specialist change-detection model ChangeMamba [3] and MLLM-based referring-segmentation models LISA [19] and PSALM [53], all fine-tuned on our dataset. As shown in Figure 10, ChangeMamba attains the highest IoU (32.41%) as a task-specific model trained on a fixed target ("new buildings"). Among MLLM-based methods, PSALM (26.93%) outperforms LISA (13.85%). DVLChat reaches 29.06% IoU, within 3.35% of the specialist. The qualitative results show DVLChat's tighter alignment with the ground truth than LISA/PSALM, especially around building boundaries and the spatial extent of new constructions.

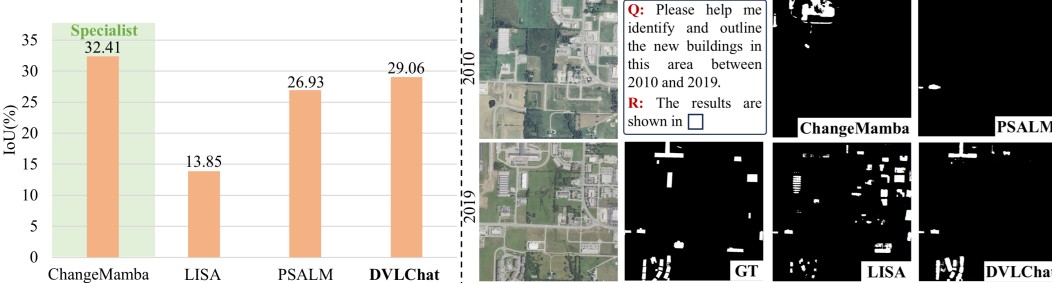

Figure 10: Performance comparison of specialist and generalist models on referring change detection.

# 6  Limitations and Future Directions

Several key directions remain for future exploration. First, DVL-Suite includes near-infrared band information from NAIP imagery, but the limited capabilities of current MLLMs in processing these spectral bands prevent their potential utilization, particularly for economic assessment tasks. Second, while DVLChat provides a unified baseline, it does not yet leverage pixel-level segmentation data to enhance numerical quantification across tasks. Finally, while DVLChat outperforms existing open-source models on most metrics, it still falls behind commercial models. Future work will focus on developing specialized algorithms and scaling up parameter size to bridge this performance gap.

# 7  Conclusion

In this paper, we present DVL-Suite, a large-scale vision-language benchmark for analyzing long-term urban dynamics through remote sensing imagery. Featuring 14,871 high-resolution multi-temporal images across 42 U.S. cities with detailed annotations spanning six urban understanding tasks, DVL-Suite enables systematic evaluation of MLLMs' capabilities from pixel-precise change detection to comprehensive temporal reasoning. Through extensive evaluation of 18 state-of-the-art models, we reveal critical insights: current MLLMs struggle significantly with long-term temporal understanding and quantitative analysis, while scaling model parameters alone proves insufficient without domain-specific training data. To address these limitations, we introduce DVL-Instruct, a specialized instruction-tuning dataset, and develop DVLChat as a baseline model that demonstrates substantial improvements, showcasing the potential of domain-specific data for advancing multi-temporal urban understanding capabilities.

# Acknowledgements

This work was supported in part by the Council for Science, Technology and Innovation (CSTI) and the Cross-ministerial Strategic Innovation Promotion Program (SIP) "Development of a Resilient Smart Network System against Natural Disasters" (funding agency: NIED), KAKENHI (25K03145). This work was also supported by NVIDIA Academic Grant. This work used computational resources on the Miyabi supercomputer, provided by The University of Tokyo through the Joint Usage/Research Center for Interdisciplinary Large-scale Information Infrastructures and High Performance Computing Infrastructure in Japan (Project ID: jh250017). Weihao Xuan is supported by RIKEN Junior Research Associate (JRA) Program.

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
