# OpenReview forum: "DynamicVL: Benchmarking Multimodal Large Language Models for Dynamic City Understanding"
_NeurIPS.cc/2025/Datasets_and_Benchmarks_Track — NeurIPS 2025 Datasets and Benchmarks Track poster_

### Official Review · Reviewer_vVPe · 2025-06-26

**Rating:** 5
**Confidence:** 5

**Summary:**

This paper introduces DVL-Suite, a multi-temporal vision-language and semantic segmentation benchmark. DVL-Suite addresses a critically important problem: multi-temporal semantic change detection and the corresponding use of vision-language models for dynamic urban understanding. Based on DVL-Suite, the authors developed a baseline model, DVLChat, which can perform VQA and referring change detection through a routing strategy. The authors extensively evaluated many state-of-the-art models, including both open-source and commercial models.

Primary Contributions

1. The paper provides an impressive large-scale multi-temporal change detection dataset with corresponding VQA pairs, which can be used to analyze specialized segmentation models and current mainstream vision-language models. Through extensive experiments, the authors demonstrate the broad applicability of this dataset.
2. The authors comprehensively analyzed existing state-of-the-art models, ranging from closed-source o4-mini to large-scale open-source models like InternVL3 78B and Qwen2.5-VL 72B. The results reveal that current models have significant limitations in remote sensing change understanding.
3. The authors developed an interesting baseline to guide future research on how to use this dataset, integrating the proposed referring change detection and VQA capabilities.

**Additional Feedback:**

Other minor weaknesses:
1. **More benchmarks in Table 1**: Table 1 only covers vision-language benchmarks. Adding comparisons with semantic change detection datasets would better highlight the dataset's impact.
2. Please fix the following typos:
   - Sometimes use "Fig." and sometimes "Figure." This needs to be consistent. In the supplementary material, it should be "border impact" rather than "broad impact."
   - Table 1 uses [BSC], [CRE], [RS] while main text uses [BCA], [CSE], [RCD] for the same tasks.
   - Page 2 has duplicate "5)" numbering and claims "seven capabilities" but only describes six tasks.
   - Figure 11 caption references "InternVL" instead of "InternVL3".
   - Claims "17 models" in methodology but "15 models" in conclusion section. The correct number is 17 models including referring segmentation models.

**Dataset Code Accessibility:**

Yes

**Dataset Code Comments:**

The authors provide detailed evaluation code and README. These files include how to apply this dataset for evaluating vision-language models, including prompt libraries and core inference modules.

**Ethical Comments:**

The authors' data is based on a publicly available data platform, NAIP, and they conducted manual annotation work. This is a common practice in the field, and the entire process does not involve any ethical concerns.

**Ethical Considerations:**

No, there are no or only very minor ethics concerns

**Final Justification:**

Thanks for the thorough reply, it resolves my earlier concerns. The paper remains a solid contribution, and my rating stays at accept.

**Limitations Weaknesses:**

1. **Clarification on NAIP Data Selection**: The authors should provide clarification on why the data only covers the United States. The authors clearly have the capability to extend the dataset to other regions and then give more significant contributions to the field for global-level semantic change detection. I'm wondering if it's due to the license or other reasons.
2. **Economic Assessment**: It is well-known in remote sensing that economic assessment is difficult to predict and judge using RGB images alone. The necessity of this task here should be clarified.
3. **More Specialized Models**: Specialized models like EarthDial [1] can also perform multi-temporal understanding and should be included in the evaluation.
4. **Potential Applications**: The authors should discuss potential applications of DVL-Suite, such as how it can be used to improve urban planning, disaster response, or environmental monitoring. This would help readers understand the practical implications of the dataset and its broader impact on society.

[1] Soni, Sagar, et al. "Earthdial: Turning multi-sensory earth observations to interactive dialogues." Proceedings of the Computer Vision and Pattern Recognition Conference. 2025.

**Strengths Contributions:**

1. **Significant Contribution of DVL-Suite**: DVL-Suite is a large-scale long-term semantic change detection and VQA dataset. This is a very valuable dataset for both developing semantic change detection models and remote sensing VLMs, filling a crucial gap in the field for long-term time-series data. Building upon TeoChatlas and other works that primarily focus on bi-temporal tasks, this work further advanced into long-term temporal sequence exploration, which is promising. Additionally, a particularly impressive aspect is that the authors created new annotations from scratch and designed VQA based on these new annotations. Currently, vision-language datasets in remote sensing are usually built upon existing datasets, whereas DVL-Suite independently collected and annotated data. This is a significant contribution and facilitates future work in integrating DVL-Suite into other remote sensing benchmarks for long-term sequence understanding tasks.
2. **Comprehensive Benchmark Results**: Another aspect that surprised me is that the authors tested a wide range of vision-language models, including o4-mini, GPT4.1, and the latest InternVL3 78B. Given the considerable performance gap between commercial VLMs and open-source models, the authors clearly demonstrated that even current SOTA commercial models cannot solve this problem, making the dataset design even more promising.
3. **Introduction of DVLChat**: Although DVLChat is far from novel, considering this is a benchmark paper, the authors' proposal of this framework makes the practical importance of the entire benchmark more apparent. That is how to integrate segmentation and VQA in future VLM designs for more detailed and versatile urban dynamic understanding. This is meaningful and inspiring for the field's methodological design, particularly for remote sensing-specific VLM development, where pixel-level understanding is crucial but most existing benchmarks do not consider pixel-level understanding.
4. **Clear and Accessible Writing**: The entire paper is well-written and flows smoothly, with detailed explanations of each task in different aspects of dynamic understanding.

---

> ### Author Rebuttal · Authors · 2025-07-30
>
> Thank you very much for recognizing DVL-Suite as a significant contribution, **filling the long-term temporal understanding gap in remote sensing**, our **comprehensive evaluation** demonstrating challenges for SOTA models, DVLChat's **meaningful integration of segmentation and VQA capabilities**, and **the clear presentation** of our work! Below, we address your questions one by one:
>
> > Q1: NAIP Data Selection.
>
> Thank you for raising insightful questions regarding our data source selection! When initially designing this study, we considered establishing the entire benchmark on a global scale. We conducted research and systematically evaluated multiple high-resolution satellite data sources with the following results:
>
> | Data Source      | Time Span        | Temporal Depth                 | Resolution  | Geographic Scope                                          | License           |
> | -| -| -| - | -| -|
> | WorldView/GeoEye | 2001–present     | Continuous                     | 31-60 cm    | Global                                                    | Commercial        |
> | Pléiades/Neo     | 2012–present     | Continuous                     | 30-70 cm    | Global                                                    | Commercial        |
> | SPOT-6/7         | 2012–present     | Continuous                     | 1.5 m       | Global                                                    | Commercial        |
> | PlanetScope/Dove | 2014–present     | Continuous                     | 3-4 m       | Global                                                    | Commercial        |
> | SkySat           | 2013–present     | Continuous                     | 50 cm       | Global                                                    | Commercial        |
> | RapidEye         | 2009–2020        | Static        | 5 m         | Global       | Commercial        |
> | Jilin-1          | 2015–present     | Continuous       | 0.5-0.75 m  | Global        | Commercial        |
> | Gaofen-2         | 2014–present     | Programmatic     | 0.8 m       | Limited access     | Policy-managed    |
> | SpaceNet         | 2016–2025        | Static   | Varied      | Multiple cities, limited temporal continuity per location | CC BY-SA 4.0      |
> | **NAIP**         | **2003–present** | **about 20 years, continuous** | **0.3-1 m** | **Continental US (9.8M km²)**                             | **Public Domain** |
>
> Based on our investigations, we discovered that only NAIP simultaneously satisfies the criteria of **being publicly accessible and redistributable**, **having a long temporal span (preferably above 10 years)**, **maintaining high resolution and data quality**, and **providing sufficient coverage with adequate diversity**. Within this geographic scope, we strategically selected 42 major cities distributed across the continental United States (as shown in Figure 4) to maximize regional diversity, covering different climatic zones, urban development patterns, and geographic contexts from coastal to inland metropolitan areas. We aim to provide the entire research community with publicly accessible, broadly covered, and high-quality data, making NAIP the most suitable for this purpose. Although SpaceNet was within our initial consideration, we found that it lacks consistent temporal coverage at the same locations, resulting in insufficient data for tracking changes, which does not meet the requirements for understanding urban dynamic changes. Additionally, we initially considered incorporating existing semantic change detection datasets such as SECOND \[1] and Hi-UCD \[2] as part of the overall data collection. However, since they only provide bi-temporal data, it remains isolated from the broader research questions we aim to address. Therefore, we collected long-term dataset from NAIP from scratch and conducted professional annotation across multiple time points, which required high investment in manual annotation costs but enables the comprehensive long-term analysis that existing datasets cannot support. This rigorous selection process ensures our benchmark provides the optimal foundation for long-term urban dynamics analysis.&#x20;
>
> We will update these considerations to better position our paper in the revised version.
>
> \[1] Yang, K., Xia, G. S., Liu, Z., Du, B., Yang, W., Pelillo, M., & Zhang, L. Asymmetric siamese networks for semantic change detection in aerial images. *TGRS 2021.*
>
> \[2] Tian, S., Zhong, Y., Zheng, Z., Ma, A., Tan, X., & Zhang, L. Large-scale deep learning based binary and semantic change detection in ultra high resolution remote sensing imagery: From benchmark datasets to urban application. *ISPRS P\&RS 2022.*
>
> > Q2: The necessity of economic assessment data.
>
> Thank you for this insightful question regarding the necessity of the Economic Assessment task. We acknowledge the challenges of economic assessment using remote sensing imagery, but we also found research has demonstrated its feasibility and significance. **Studies have consistently shown that satellite imagery can effectively predict economic indicators when combined with machine learning approaches**, from pioneering work on poverty prediction \[1] to more recent advances in understanding economic well-being across different regions \[2,3]. This body of research establishes the scientific foundation for such applications in remote sensing analysis. Furthermore, our dataset is based on NAIP imagery, which actually includes near-infrared (NIR) bands in addition to RGB channels. The NIR data could largely enhance the ability to distinguish between farmland, forests, water bodies, and urban structures, enabling more accurate tracking of land use changes and assessment of regional economic development patterns. As a Dataset and Benchmark contribution, our primary goal is to present comprehensive evaluation tasks that expose current limitations and guide future research directions. While we focus on RGB channels to maintain compatibility with existing multimodal models, the inclusion of Economic Assessment tasks reveals important gaps in current MLLMs' capabilities and establishes a foundation for future work that could leverage additional spectral information. We will incorporate this discussion and additional references into our revised manuscript to better justify the inclusion of this task and clarify our benchmark design goal.
>
> \[1] Jean, N., Burke, M., Xie, M., Alampay Davis, W. M., Lobell, D. B., & Ermon, S. Combining satellite imagery and machine learning to predict poverty. *Science.*
>
> \[2] Yeh, C., Perez, A., Driscoll, et al. Using publicly available satellite imagery and deep learning to understand economic well-being in Africa. *Nature communications.*
>
> \[3] Pettersson, M. B., Kakooei, M., Ortheden, J., Johansson, F. D., & Daoud, A. Time Series of Satellite Imagery Improve Deep Learning Estimates of Neighborhood-Level Poverty in Africa. *IJCAI.*
>
> > Q3: More models.
>
> Thank you for this valuable suggestion regarding the inclusion of specialized models in our evaluation. We conducted additional experiments comparing EarthDial with general vision-language models and present the results below:
>
> | Models | MCQ  | BCA-Report | CSE-Report | Regional | Dense |
> | - | - | - | - | - | - |
> | Qwen2.5-VL 3B    | 24.7 | 2.99       | 1.72       | 2.76     | 2.38  |
> | EarthDial 4B RGB | 30.3 | 1.10       | 1.03       | 1.53     | 0.90  |
>
> **While EarthDial demonstrates competitive performance on multiple-choice questions (30.3% vs 24.7%), it shows significant limitations in open-ended generation tasks, with substantially lower scores across all report and captioning metrics**. This comparison highlights a critical limitation of current domain-specific models: they excel at constrained classification tasks but struggle with comprehensive language generation and reasoning required for complex urban dynamics understanding. These results reinforce our argument that advancing general vision-language models through specialized datasets like DVL-Suite is more promising than developing domain-specific architectures, as the former can maintain both analytical precision and natural language interaction capabilities essential for practical urban analysis applications. We will include this analysis and comparison in our revised manuscript to provide a more comprehensive evaluation.
>
> > Q4: Potential applications.
>
> Thank you for your advice about highlighting the potential applications of our developed benchmark. DVL-Suite's unique combination of long-term temporal coverage (18 years), high-resolution imagery (1024×1024), and multi-level analysis capabilities positions it as a valuable tool for urban planning, disaster response, and environmental monitoring applications. Our framework enables comprehensive analysis of historical development patterns, infrastructure vulnerability assessment, and urban sustainability challenges including heat island effects and green space development.
>
> Importantly, the vision-language model paradigm offers advantages over **traditional segmentation-then-analysis approaches by enabling natural language interactions that make complex remote sensing analysis accessible to policymakers and the general public without specialized remote sensing expertise**. This democratization of urban analysis capabilities allows for more inclusive decision-making processes and evidence-based monitoring and policy development. We will include these claims in the revised manuscript.
>
> > Additional Feedback: Minor weaknesses.
>
> Thank you for these very insightful suggestions and your meticulous attention to detail! We greatly appreciate your thorough review that identified these points for improvement. We will address all of these corrections in our revised manuscript, including expanding Table 1 to incorporate semantic change detection dataset comparisons as suggested, ensuring consistent figure referencing throughout, and correcting the model count discrepancies.

---

> > ### Comment · Reviewer_vVPe · 2025-08-04
> >
> > Thanks for the thorough reply, it resolves my earlier concerns. The paper remains a solid contribution, and my rating stays at accept.

---

> > > ### Author Response · Authors · 2025-08-04
> > >
> > > Thank you very much for acknowledging our contributions and rebuttal! We will incorporate your suggestions into the manuscript to further strengthen the contributions of our work. We sincerely appreciate your valuable feedback and the time you dedicated to reviewing our paper.

---

### Official Review · Reviewer_xvx3 · 2025-06-30

**Rating:** 6
**Confidence:** 5

**Summary:**

DynamicVL proposes DVL-Suite, the first large-scale benchmark designed to test and improve multimodal large language models for long-term urban change understanding based on remote sensing imagery. The suite brings together 15,063 1-meter resolution remote sensing image stacks covering 42 US megacities from approximately recent 20 years and splits them into: (i) DVL-Bench, with seven tasks covering pixel-level change detection, quantitative rate of change estimation, regional captioning, and dense temporal narratives; and (ii) DVL-Instruct, a set of instructions tuned specifically for multi-temporal Earth observation inference. Benchmarking 17 state-of-the-art open source and commercial MLLMs shows that even the best model only achieves a low overall accuracy, highlighting the poor performance of current models in temporal reasoning and numerical precision. Using DVL-Instruct, the authors designed DVLChat, a LoRA-enhanced task routing baseline model that can jointly perform image-level question answering and pixel-level segmentation. DynamicVL establishes a comprehensive, multi-granular benchmark and strong baseline to promote progress in long-term remote sensing understanding.

**Dataset Code Accessibility:**

Yes

**Dataset Code Comments:**

The dataset is fully accessible, with a transparent and logically structured codebase.

**Ethical Comments:**

The data generation process complies with ethical guidelines. Using VLM to understand urban expansion does not have negative implications for society.

**Ethical Considerations:**

No, there are no or only very minor ethics concerns

**Final Justification:**

The responses addressed my potential worries. So I maintain my original score.

**Limitations Weaknesses:**

1. Figure 2 mentions that after the dataset is human annotated, GPT is used to generate QA pairs and the entire report, but there may be illusions in this process. How did the author conduct quality control throughout the process? In addition, who is responsible for the supervisor role in the claim section 3, professionals or annotators?
2. It is difficult to support the claim of effectiveness of DVL-Instruct by fine-tuning only one model (Qwen2.5-VL 7B). The authors should compare with at least one other model, such as InternVL3 8B.
3. The author mentioned that "the original LISA model was unable to perform referring segmentation while maintaining high open-ended capabilities due to optimization conflicts". However, there are not detailed results in the paper. More details on this issue are needed for understanding the motivation of the DVLChat's design.

**Strengths Contributions:**

1. Scale and originality: DVL-Suite introduces a novel large-scale dataset that integrates fine-grained pixel-level change detection with comprehensive vision-language annotations across largely extended temporal sequences.
2. Well-designed tasks: The proposed series of tasks can evaluate the capabilities of vision-language models in long-term time series understanding. The tasks are well thought out and cover real-world applications.
3. Extensive experiments: Detailed results of different model families, model sizes, closed source and open source models on the DVL-Bench.
4. Good presentation: The paper is well-structured, with clear explanations of the dataset, tasks, and experimental results.

---

> ### Author Rebuttal · Authors · 2025-07-30
>
> Thank you very much for highly recognizing **the scale and originality of our DVL-Suite dataset**, **the well-designed tasks** that evaluate long-term time series understanding for real-world applications, our **extensive experiments across diverse model families**, and **the clear presentation** of our work! We really appreciate your positive feedback and respond to each point:
>
> > Q1.1: Quality control; Q1.2: Responsible person.
>
> Thank you for raising this question about quality control in our annotation pipeline. As described in Section 3, we implement a comprehensive three-stage quality control process to ensure data reliability: self-examination, cross-examination to correct errors (including false labels, missing details, and inaccurate captions), and supervisor review of randomly sampled annotations. The supervisors are remote sensing experts with graduate-level knowledge who verify the accuracy and consistency of **both the initial mask annotations by human annotators and subsequent LLM-generated instruction pairs**, ensuring that all content maintains factual correctness and domain expertise. We conducted rigorous supervisor review on 1,000 randomly sampled annotations, with any unqualified annotations returned for refinement. This quality control process effectively maintained high annotation quality and data reliability throughout the entire dataset creation pipeline, from initial semantic segmentation to final instruction-answer pairs. We will include a more detailed description of these quality control measures and supervisor qualifications in the revised manuscript.
>
> > Q2: More experiments.
>
> Thank you for raising the concern regarding the generalizability of DVL-Instruct's effectiveness. We have conducted additional experiments across multiple model architectures and sizes to validate our approach. The table below compares zero-shot performance with DVL-Instruct fine-tuned results across both Qwen2.5-VL and InternVL3 families, where all metrics except cIoU (segmentation) represent average scores.
>
> | Models                  | MCQ  | BCA-Report | CSE-Report | Regional | Dense | cIoU (Seg) |
> | ----------------------- | ---- | ---------- | ---------- | -------- | ----- | ---------- |
> | Qwen2.5-VL 3B           | 24.7 | 2.99       | 1.72       | 2.76     | 2.38  | -          |
> | DVLChat 3B (Qwen2.5-VL) | 31.6 | 3.43       | 2.25       | 3.78     | 3.19  | 26.9       |
> | Qwen2.5-VL 7B           | 23.3 | 2.94       | 1.73       | 3.21     | 2.85  | -          |
> | DVLChat 7B (Qwen2.5-VL)  | 33.3 | 3.47       | 2.51       | 3.98     | 3.40  | 29.1       |
> | InternVL3 8B            | 23.9 | 2.99       | 2.15       | 3.69     | 2.97  | -          |
> | DVLChat 8B (InternVL3)   | 37.6 | 3.65       | 2.41       | 3.83     | 3.42  | 30.7       |
> | InternVL3 14B           | 27.2 | 3.02       | 2.36       | 3.96     | 3.22  | -          |
> | DVLChat 14B (InternVL3)  | 41.3 | 3.68       | 2.61       | 3.99     | 3.48  | 31.3       |
> | Qwen2.5-VL 32B          | 31.4 | 3.04       | 2.60       | 3.90     | 2.91  | -          |
> | DVLChat 32B (Qwen2.5-VL) | 42.1 | 3.60       | 2.65       | 4.22     | 3.68  | 34.7       |
>
> **The results demonstrate consistent and substantial improvements across both model families and various sizes**. DVL-Instruct fine-tuning consistently enhances performance across all metrics, with particularly notable gains in MCQ accuracy (6.9-10.7% improvements) and captioning tasks. This validates that DVL-Instruct's effectiveness generalizes beyond a single model architecture, confirming its utility for advancing not only high-level reasoning but also fine-grained segmentation capabilities in multi-temporal urban understanding. We will add these additional experiments and the detailed analysis to the revised version.
>
> > Q3: Examples of optimization conflicts between VQA and referring change detection.
>
> Thank you for this insightful clarification request regarding the optimization conflicts! Through our preliminary experiments, we observed several critical issues when directly applying LISA to our multi-temporal urban understanding tasks. Specifically, the model exhibited significant instruction-following failures across different task types: (1) for single-choice questions, LISA frequently generated invalid responses like "O." instead of selecting from valid options; (2) for multiple-choice questions, it produced nonsensical outputs such as "O, 2, 3, 5, 6, 7, 8, 9, 10, 11, 12, 13, 14, 15, 16, 17, 18, 19," rather than coherent answer combinations; and (3) most critically, for open-ended reporting tasks, the model consistently defaulted to segmentation-style responses like "Sure, it is \[SEG]" instead of generating descriptive analysis. These failures stem from the optimization conflict between segmentation and language generation objectives within the original unified training framework. The segmentation branch's dominance during joint optimization interferes with the model's ability to maintain proper instruction-following behavior for diverse question types. Our DVLChat design addresses this by implementing task-specific LoRA modules with explicit routing mechanisms, effectively isolating the optimization paths for question-answering and segmentation tasks. This architectural separation prevents task interference while enabling the model to excel at both multi-temporal urban analysis and precise change detection within a single framework. We will include these specific examples in our revised manuscript to more clearly demonstrate the motivation and effectiveness of our design approach.

---

> > ### Comment · Reviewer_xvx3 · 2025-08-03
> >
> > I appreciate the authors' detailed response. The authors' rebuttal has taken care of my potential worries. This work makes a good contribution. My score remains unchanged (strong accept).

---

> > > ### Author Response · Authors · 2025-08-03
> > >
> > > Thank you so much for your positive feedback and for recognizing our contribution! We're committed to implementing your suggestions in our revision. We are very grateful for the time you took to review and respond to our rebuttal.

---

### Official Review · Reviewer_8uLB · 2025-07-02

**Rating:** 5
**Confidence:** 4

**Summary:**

This paper introduces DVL-Suite, a novel remote sensing dataset for long-term time series visual language understanding. The authors collected and manually labeled NAIP data to create DVL-Suite, which contains about 15,000 images. DVL-Suite is a versatile dataset. From the paper, its main functions include evaluating vision-language models by 1) multiple-choice questions for change detection; 2) open ended questions for change detection, including basic change analysis and change speed estimation; 3) economic assessment, including population density, nighttime light intensity, CO2 and O3 emissions, and temperature; 4) regional captioning; 5) dense temporal captioning. It evaluates the multi-temporal capabilities of change detection models. Based on DVL-Suite, the authors tested many current vision language models, such as OpenAI's o4-mini and GPT-4.1. The results show that existing models have difficulty understanding changes over time. To address this problem, the authors developed DVLChat, a baseline model that can both do change detection and answer questions.

**Additional Feedback:**

Please address the questions and weaknesses. My final score will be based on the authors‘ response.

**Dataset Code Accessibility:**

Yes

**Dataset Code Comments:**

The authors provide complete data and test code, clearly explaining how to use vLLM to evaluate the entire dataset.

**Ethical Considerations:**

No, there are no or only very minor ethics concerns

**Final Justification:**

The authors' additional experiments could support their claims. Also, the compatibility proves that the proposed suite has broad downstream potential. Therefore, the authors' rebuttal further strengthens their contributions. I think it's a good work that will have broad impact, so I raise my score to 5 and recommend acceptance.

**Limitations Weaknesses:**

1. In the paper, it mentions that scaling parameters will significantly improve the QA and captioning scores. Will there be better results for the model fine-tuned using DVL-Instruct? Please add experiments to detail this point.
2. From Figure 7, it seems that the mask category of DVL-Suite is similar to the SECOND dataset. You should provide a detailed description on this. Please discuss the feasibility of more downstream tasks, such as Domain Adaptation?
3. The paper indicates that the change analysis and change speed evaluation reports are based on pixel-level annotations, with results presented to two decimal places according to the provided data. Given that segmentation masks typically contain inherent labeling errors, it's important to clarify how to address this issue in their evaluation methodology to ensure reliable quantitative assessments.
4. The main body lacks a detailed description of environmental assessment. It would be better to move this information from supplementary materials to the main text.
5. More discussion of the distinctions between this work and other remote sensing VLM benchmarks [1] would help readers better understand the specific contributions of DVL-Suite. [1] Teochat: A large vision-language assistant for temporal earth observation data.

**Strengths Contributions:**

1. The proposed data DVL-Suite fills a critical gap by introducing the first comprehensive benchmark for long-term urban dynamics understanding.
2. The paper introduces multi-image/video-like tasks in remote sensing, which can bridge the gap between the generic CV field and the remote sensing field.
3. The paper provides details to construct the entire benchmark, including data collection, quality check, annotation, and prompts for QA generation and evaluations.
4. The proposed baseline DVLChat has superior performance compared with other open-source VLMs and have the ability to do segmentation.

---

> ### Author Rebuttal · Authors · 2025-07-30
>
> Thank you for recognizing that our **DVL-Suite fills a critical gap** as the first comprehensive benchmark for long-term urban dynamics understanding, our introduction of the **designed tasks that bridge generic CV and remote sensing fields**, the **detailed construction methodology**, and **our DVLChat baseline's superior performance** with segmentation capabilities! Here, we address your concerns in detail:
>
> > Q1: Scaling parameters.
>
> Thank you for raising this insightful question about parameter scaling effects with DVL-Instruct fine-tuning. We conducted additional experiments using Qwen2.5-VL as the base architecture across different model sizes (3B, 7B, 32B), and the results clearly demonstrate consistent improvements with parameter scaling. To reflect the overall performance on each task, we report the average score for each category, except for the cIoU metric used in referring change detection.
>
> | Models      | MCQ  | BCA-Report | CSE-Report | Regional | Dense | cIoU (Seg) |
> | ----------- | ---- | ---------- | ---------- | -------- | ----- | ---------- |
> | DVLChat 3B  | 31.6 | 3.43       | 2.25       | 3.78     | 3.19  | 26.9       |
> | DVLChat 7B  | 33.3 | 3.47       | 2.51       | 3.98     | 3.40  | 29.1       |
> | DVLChat 32B | 42.1 | 3.60       | 2.65       | 4.22     | 3.68  | 34.7       |
>
> As shown in the supplementary results, our DVLChat models exhibit significant performance gains across all evaluation metrics when scaled from 3B to 32B parameters. Specifically, the multiple-choice QA accuracy improves from 31.6% to 42.1%, captioning tasks show steady improvements (Regional: 3.78 to 4.22, Dense: 3.19 to 3.68), and referring segmentation cIoU increases from 26.9% to 34.7%. **This scaling behavior confirms that larger models can better leverage the specialized knowledge encoded in DVL-Instruct**, demonstrating the effectiveness of our instruction-tuning approach for dynamic urban understanding tasks. We will include the detailed experimental results in the revised version.
>
> > Q2: Compatibility with other change detection datasets.
>
> Thank you for the insightful and careful observation regarding the mask categories in Figure 7 and the comparison with the SECOND dataset. Our DVL-Suite follows established land cover type conventions widely adopted in remote sensing research, which creates valuable compatibility with existing datasets like SECOND. The only distinction compared to SECOND lies in our deliberate consolidation of "tree" and "low vegetation" into a unified "vegetation" class to avoid potential annotation errors in high-resolution imagery, while all other category definitions remain identical. This categorical alignment is advantageous as it naturally enables other potential tasks, such as domain adaptation studies between different datasets and geographic regions, promoting broader research applications.
>
> Moreover, DVL-Suite offers advantages in scale and temporal flexibility compared to SECOND's bi-temporal design. Our multi-temporal framework allows flexible temporal combinations (adjacent pairs, first-last frames, etc.), and for domain adaptation studies with bi-temporal datasets like SECOND, researchers can simply extract the first and last frames from our sequences to create maximum temporal differences, while maintaining our setting as a much larger, multi-temporal, and more diverse semantic change detection dataset. Although DVL-Suite is primarily designed as a vision-language benchmark, it also provides additional and prominent advantages for traditional semantic change detection tasks due to its scale and temporal richness. We will include this detailed analysis in the revised version to better highlight DVL-Suite's broader applicability and versatility for research beyond only vision-language models.
>
> > Q3: Data tolerance.
>
> Thank you for this important question regarding evaluation reliability. We fully acknowledge the inherent limitations of pixel-level annotations. However, our evaluation methodology and precision settings are designed based on several key considerations to address potential annotation challenges. As detailed in our supplementary material's evaluation prompts, our quantitative assessment methodology addresses inherent labeling errors through robust evaluation metrics that focus on overall accuracy rather than pixel-perfect precision. **For Change Quantification Accuracy (BCA reports), we evaluate the alignment between predicted and ground truth transition percentages and completeness of quantitative information**, while for Change Rate Precision (CSE reports), we assess how accurately the predicted percentage values match the actual change rates described in ground truth results. Moreover, given our data's 1-meter GSD and 1024×1024 pixel images covering over 1 square kilometer, even a 0.01% area change corresponds to more than 100 square meters of actual land area, which is crucial for small-scale land use change applications. Therefore, maintaining two decimal precision is essential for capturing these meaningful, subtle changes, and our evaluation approach enables reliable relative assessment of different models' quantitative reasoning capabilities by ensuring all algorithms are evaluated under the same conditions and annotation standards. We will incorporate this analysis and explanation in the revised version.
>
> > Q4: The position of the data source of the environmental assessment.
>
> Thank you for highlighting the environmental assessment component. We believe our current organization effectively balances comprehensive content presentation with space constraints inherent to conference submissions. The environmental assessment task encompasses multi-source Earth observation data, including ozone density, nighttime light intensity, temperature, and population measurements. Our methodology first acquires these environmental datasets (detailed sources provided in Appendix D), then spatially aligns them with our remote sensing imagery through resampling and interpolation. To ensure data accuracy, we only select regions with complete temporal and spatial coverage across all environmental products, excluding areas where certain products lack coverage despite performing interpolation. In the revised version, we will increase the prominence of environmental assessment discussions in the main text while directing readers to supplementary materials for detailed technical specifications.
>
> > Q5: More discussions about the differences between TEOChat and DVL-Suite.
>
> Thank you for this question regarding the distinctions between DVL-Suite and TeoChatlas. As summarized in Table 1, our work differs from TeoChatlas in three fundamental aspects. First, TeoChatlas focuses on short-term analysis with only 2.07 average temporal frames, while our **DVL-Bench and DVL-Instruct feature&#x20;**&#x61;n **average of about 6.9 frames** spanning 18 years (2005-2023), enabling comprehensive long-term urban dynamics analysis. Second, TeoChatlas employs low-resolution imagery (224×224 pixels) that constrains geographic detail analysis, whereas **our high-resolution imagery (1024×1024 pixels) captures much broader and fine-grained urban elements essential** for detailed change detection. Finally, TeoChatlas primarily repurposes existing remote sensing annotations, while **our DVL-Suite introduces newly collected and annotated data with a comprehensive task taxonomy for dynamic city understanding**. Our framework creates novel multi-temporal vision-language tasks that advance multimodal AI capabilities while aligning with widely used semantic change detection categories. This design ensures compatibility with both emerging multimodal foundation models and existing remote sensing methods, establishing a more comprehensive benchmark for urban dynamics understanding. We will incorporate this detailed comparison into our revised manuscript to better highlight these key distinctions.

---

> > ### Comment · Reviewer_8uLB · 2025-08-02
> >
> > Thank you for the additional experiments and analysis. After checking all reviews and replies, my concerns are fully addressed. I appreciate the value of DynamicVL in advancing both vision-language models and change detection in remote sensing, so I will raise my score to 5 and recommend acceptance.

---

> > ### Author Response · Authors · 2025-08-02
> >
> > Thank you very much for your positive response to our additional experiments and analysis! Your constructive feedback has clearly helped us in strengthening our contributions. We sincerely appreciate the time and effort you have invested in this review process.

---

### Official Review · Reviewer_qr7v · 2025-07-03

**Ethics Flags:** Data privacy, copyright, and consent,…
**Rating:** 4
**Confidence:** 5

**Summary:**

The paper introduces DVL-Suite, a comprehensive benchmark for evaluating Multimodal Large Language Models (MLLMs) on long-term urban dynamics analysis using remote sensing imagery. This addresses a critical gap in multi-temporal Earth observation research.  In addition, the proposed dataset is extensive and well-annotated, covering diverse tasks from pixel-level change detection to scene-level urban narratives.

**Additional Feedback:**

See the above.

**Dataset Code Accessibility:**

Partly

**Dataset Code Comments:**

1. Although the dataset itself has been released, none of the model outputs or corpora are public.

2. The code for the evaluation indicators is not public, and there is a lack of a complete testing process. An automated code implementation should be provided.

**Ethical Comments:**

1. The selected LLM may introduce potential biases related to remote sensing visual question answering (VQA).

2. A dedicated section discussing ethical constraints should be included, detailing the dataset’s origin, the specific usage permissions or licenses under which it was obtained, and the steps taken to ensure responsible and compliant use of the data.

**Ethical Considerations:**

No, there are no or only very minor ethics concerns

**Final Justification:**

Like other reviewers, **I acknowledge that this paper makes a valuable contribution by releasing a large-scale, multi-temporal change detection dataset, thereby broadening the scope of available resources. It also provides extensive benchmarking of large language models (LLMs), which is a meaningful addition to the field**. However, the analysis of the proposed DVLChat system remains insufficient, even after the rebuttal, and the ablation experiments on various strategies do not demonstrate significant performance improvements. Moreover, DVLChat was tested across multiple established remote sensing benchmarks beyond Qwen2.5:7B VL—this is a valuable and extensive set of experiments that should be emphasized more prominently in the final version.

**That said, I believe a thorough revision would be challenging if the paper were accepted in its current form, given the need for additional experiments, deeper analysis, and the provision of data and code resources.** While the authors have addressed some of my concerns during the rebuttal phase, several important issues—particularly Q1, Q9, and the analysis of DVLChat—remain unresolved**. For this reason, I remain cautious about the potential for substantial improvement in future iterations of this work. Considering the paper’s initial quality, the generally positive feedback from other reviewers, and the authors’ rebuttal performance, I have raised my score to near acceptance rather than full acceptance.

**Limitations Weaknesses:**

1. While the paper reports extensive results, it does not sufficiently explain why certain models (e.g., commercial ones like GPT-4.1) outperform others. Are the observed differences attributable to architectural variations, training data diversity, or optimization strategies? Consider adding ablation studies (e.g., the impact of temporal attention mechanisms) or attention visualizations to clarify the sources of performance gaps.

2. The dataset is limited to U.S. cities, which constrains the generalizability of the findings to other regions (e.g., rapidly urbanizing areas in developing countries). Furthermore, potential biases—such as seasonal variations in imagery—are not addressed.

3. Based on the reported results, DVLChat still lags behind commercial models (e.g., GPT-4.1). The paper does not explore whether scaling model capacity, enhancing multi-modal fusion, or incorporating pixel-level segmentation could help bridge this gap.

4. Key training details (e.g., hyperparameters, optimizer configurations) are relegated to the appendix, which may hinder reproducibility. It would be beneficial to surface essential settings in the main text.

5. None of the model outputs or corpora are publicly available, although the datasets themselves have been released. Releasing representative results would strengthen transparency.

6. The evaluation metrics are not sufficiently explained and are difficult to follow. A detailed description should be provided in the appendix.

7. Several figures (e.g., the Sankey diagrams in Figure 4) have small text and low color contrast, making the visual trends hard to interpret.

8. The paper overlooks established approaches in multi-temporal remote sensing (e.g., LSTM-based change detection methods), which could help situate the novelty of MLLMs in a clearer context.

9. The details of the dataset simulation process, particularly the procedures for constructing the text descriptions, are insufficiently described. It is unclear which portions of the descriptions were manually authored and which were generated by GPT-based models. Furthermore, the rationale for selecting these specific LLMs should be clearly articulated. The authors should explain how the accuracy of the generated text descriptions in the training set was verified and provide evidence or justification demonstrating that the chosen LLMs possess sufficient knowledge relevant to remote sensing visual question answering (VQA).

10. Terms such as “multi-temporal” and “long-term” are used inconsistently. Consider unifying the terminology (e.g., consistently using “multi-temporal long-term”) to avoid confusion.

**Strengths Contributions:**

1. The dataset fills the gap in the field.

2. The authors tested a wide range of large models.

3. The organization of the paper is good and makes it easier to understand tasks and answers.

---

> ### Author Rebuttal · Authors · 2025-07-30
>
> Thank you for recognizing that our **dataset fills a gap in the field**, our **comprehensive evaluation across a wide range of large models**, and **the clear organization** that facilitates understanding of the tasks and methodologies! Below, we address your concerns one by one:
>
> > Q1: Deeper analysis of the commercial models' results.
>
> Thank you for your questions regarding model performance differences and suggestions for deeper analysis.
>
> Here, we would clarify that the suggestion to conduct ablation studies on commercial models such as GPT-4.1 and o4-mini faces fundamental technical limitations. **These well-performing commercial models are black-box with inaccessible architectural details, attention mechanisms, or training processes**, making it technically infeasible to perform ablation studies on temporal attention mechanisms or optimization strategies.
>
> In addition, we respectfully note that the suggestion to use attention visualizations may face methodological concerns. Recent findings \[1] reveal that large multimodal models consistently allocate high attention weights to irrelevant visual tokens that contribute minimally to overall performance, a phenomenon similar to attention sink phenomena observed in language models. These systematic attention biases make direct visualization potentially misleading for interpreting performance differences, as inherent attention allocation artifacts would confound architectural or training differences.
>
> \[1] Kang, S., Kim, J., Kim, J., & Hwang, S. J. See what you are told: Visual attention sink in large multimodal models. *ICLR 2025.*
>
> > Q2: NAIP data selection.
>
> Thank you for raising these concerns regarding the dataset scope and potential biases. Here, we address your concerns one by one:
>
> **Geographic Scope Justification**: While focusing on U.S. cities may appear geographically constrained, this decision followed systematic evaluation of global high-resolution satellite sources. Only NAIP simultaneously provides **public accessibility and redistributability, about 20 years of continuous temporal coverage, sub-meter resolution, and comprehensive geographic coverage (9.8M km²)**. Commercial alternatives (WorldView, Pléiades) present significant accessibility barriers, while open alternatives like SpaceNet lack sufficient temporal continuity at the same locations. Within the continental U.S., we selected 42 cities across diverse climatic zones and development patterns to maximize regional variation and ensure robust model evaluation. Additional justifications for selecting NAIP data can also be found in our response to R#vVPe.
>
> **Addressing Seasonal Diversity**: We appreciate the concern about seasonal appearance variations within each land cover class. DVL-suite contains clear seasonal diversity across the 18-year span, such as vegetation appearing in various states (lush green, autumn colors, dried grass, bare winter trees). Our annotation approach focuses on semantic land cover types (such as vegetation, buildings, water) rather than appearance-based classifications, following established practices in remote sensing datasets \[1]\[2]. This absolute semantic system enables models to learn robust cross-seasonal understanding essential for real-world deployment.
>
> We will incorporate these design rationales in the revised manuscript to better present our data selection and methodological rigor.
>
> \[1] Wang, J., Zheng, Z., Lu, X., & Zhong, Y. LoveDA: A Remote Sensing Land-Cover Dataset for Domain Adaptive Semantic Segmentation. *NeurIPS D\&B Track 2021.*
>
> \[2] Yang, K., Xia, G. S., Liu, Z., Du, B., Yang, W., Pelillo, M., & Zhang, L. Asymmetric siamese networks for semantic change detection in aerial images. *TGRS 2021.*
>
> > Q3: Bridge the gap.
>
> Thank you for your observations regarding the model performance gap. Following your suggestions, we conducted model capacity scaling experiments based on three Qwen2.5-VL variants, with results presented in the table below (all results represent average scores to reflect overall capabilities under respective tasks).
>
> | Models| MCQ | BCA-Report | CSE-Report | Regional | Dense  |
> | - | - | -| - | - | - |
> | *GPT4.1* | *32.5* | *3.02* | *2.23* | *4.46*| *3.98* |
> | DVLChat 3B  | 31.6   | 3.43 | 2.25  | 3.78| 3.19   |
> | DVLChat 7B  | 33.3   | 3.47 | 2.51| 3.98 | 3.40 |
> | DVLChat 32B | 42.1 | 3.60  | 2.65  | 4.22 | 3.68|
>
> We found that as the parameters are increased (i.e., enhanced model capacity), DVLChat's capabilities improve across all dimensions. The largest 32B model significantly outperforms GPT4.1 in average accuracy on multiple choice questions, while BCA-Report and CSE-Report scores also exceed those of GPT4.1. For regional change and dense temporal captioning tasks, although gaps remain, the 32B model demonstrates a reduced performance gap relative to GPT4.1 compared to the 7B version. This validates the effectiveness of scaling model capacity. Regarding pixel-level segmentation for overall capability enhancement, we discussed this in the original paper (Section 5.2 "Limitations"), noting that current methods face challenges in effectively integrating segmentation data with numerical quantification tasks across different temporal scales.
>
> Here, we would emphasize that as a dataset and benchmark paper, our primary contribution lies in providing the research community with comprehensive data and systematic evaluation insights, revealing the capability limitations of current state-of-the-art models in these scenarios while offering macro-level research directions for future investigators. We will incorporate this detailed analysis in the revised version.
>
> > Q4: The position of key training details; Q5, Dataset and code comment2: Model outputs or corpora; Q6: Evaluation metrics; Q7: Several figures; Q10: Consistent terms.
>
> Thank you for these suggestions. We will address these concerns in the revised version by moving essential training details to the main text, improving figure quality with better readability, releasing representative model outputs, providing more explicit evaluation metric descriptions, and establishing consistent terminology throughout the paper.
>
> > Q8: LSTM-based change detection methods.
>
> Thank you for the suggestion. While LSTM-based methods have made important historical contributions to change detection, we focused our baseline comparisons on more recent state-of-the-art approaches like ChangeMamba. To address your suggestion and provide a comprehensive context, we conducted additional comparisons with the state-of-the-art LSTM-based SiamCRNN \[1]. Additionally, we included the transformer-based SCanNet \[2] to further strengthen our evaluation, alongside scaling DVLChat to 32B parameters:
>
> | Method| cIoU |
> | - | - |
> | SiamCRNN| 27.6 |
> | SCanNet| 30.4 |
> | ChangeMamba | 32.4 |
> | DVLChat 7B | 29.1 |
> | DVLChat 32B | 34.7 |
>
> These results demonstrate that DVLChat 32B achieves superior performance compared to all specialist methods, while DVLChat 7B remains competitive with established approaches. This analysis clearly positions multimodal large language models as a promising paradigm that can surpass traditional specialist architectures while providing unified capabilities for both referring segmentation and question-answering tasks. We will incorporate this comprehensive analysis into our revised version to better contextualize our contributions within the broader field of remote sensing.
>
> \[1] Chen, H., Wu, C., Du, B., Zhang, L., & Wang, L. Change detection in multisource VHR images via deep Siamese convolutional multiple-layers recurrent neural network. *TGRS 2019.*
>
> \[2] Ding, L., Zhang, J., Guo, H., Zhang, K., Liu, B., & Bruzzone, L. Joint spatio-temporal modeling for semantic change detection in remote sensing images. *TGRS 2024.*
>
> > Q9, Ethical comment1: Detailed annotation process; the rationale for selecting specific LLMs.
>
> Thank you for this feedback. While we described our multi-stage pipeline in Section 3 and Figure 2, we recognize that additional details would be helpful. For additional clarity: human domain experts performed all basic annotations, including semantic segmentation, keyframe identification, and period-specific caption drafts, while **GPT-4.1 was used solely for language enhancement rather than creating factual content**. *Our quality control involved rigorous verification through self-examination, cross-examination, and supervisor review of 1,000 randomly sampled sets of annotations, with unqualified entries returned for human refinement (Lines 144-147).* We selected GPT-4.1 because it demonstrated superior performance in instruction diversification while maintaining consistency in technical terminology when provided with expert-drafted content. This human-expert-driven approach with LLM linguistic assistance follows established practices in specialized domain dataset construction. Since all domain-specific knowledge originates from certified remote sensing annotators rather than the LLM, accuracy verification occurred at the human expert level during the annotation process, with GPT-4.1 serving only as a supervised linguistic refinement tool. We will add these detailed descriptions in the revised version.
>
> > Ethical comment2: Data source.
>
> Thank you for raising this point about ethical considerations and data licensing. Our DVL-Suite dataset is built entirely on NAIP imagery, which is **publicly accessible government data with broad usage permissions**. We will add a dedicated ethics section detailing data sources, licensing terms, and responsible use practices in our revised version.
>
> > Dataset and code comment2: Evaluation code.
>
> Thank you for raising this concern about code availability. We would like to respectfully clarify that we have provided a complete anonymous GitHub repository at the code URL specified in our submission, which includes comprehensive evaluation code implementations. We will add the repository link directly to the abstract in the revised version.

---

> > ### Comment · Reviewer_qr7v · 2025-08-04
> > **On the Q1: Deeper analysis of the commercial models' results.**
> >
> > It is known that the commercial model is closed-source, but understanding the reasoning behind each model’s predictions is crucial for analyzing outputs, biases, and decision-making processes. This is the core issue, I am raising, the need to release model outputs or corpora to further progress in this field. In my view, this is extremely important.
> >
> > Additionally, your DVLChat is based on Qwen2.5:7b, but I have not found a clear justification for why it is superior to the baseline. It would be valuable to demonstrate this through effective ablation studies and statistical testing, ensuring that the differences are significant. I would also like to emphasize the importance of evaluating the corpus, particularly its instruction-following ability, as the model outputs rely on it. These experimental concerns are also shown in other reviewers. The benchmark track should have more complete benchmark tests
> >
> > In my opinion, the benchmark track should aim to meet or even surpass the main track in both direction and performance.

---

> > ### Comment · Reviewer_qr7v · 2025-08-04
> > **On the Q3: Bridge the gap.**
> >
> > It is meaningful to provide more scaling laws in DVChat architectures, but there is a lack of evaluation regarding the model's details, as well as whether it could outperform the baseline in other benchmarks. If not, consider prompt methods for your models to better assess the performance of your DVChat model.

---

> > ### Comment · Reviewer_qr7v · 2025-08-04
> > **The code and evaluation is incomplete.**
> >
> > "The code for the evaluation indicators is not public, and there is a lack of a complete testing process." and "An automated code implementation should be provided.", please provide them.

---

> > ### Author Response · Authors · 2025-08-06
> >
> > Thank you for your valuable time on the review and your constructive comments on improving our work! We greatly appreciate your valuable feedback, and our entire team has been dedicating all our efforts over the past two days to conduct additional experiments based on your suggestions. Here, we address your concerns one by one:
> >
> > > Q1: The model's outputs and corpora.
> >
> > Thank you for your inquiry regarding the model outputs and corpora. We have saved comprehensive outputs from all 17 evaluated models across all benchmark tasks, which will serve as valuable resources for future research and reproducibility studies. However, due to NeurIPS rebuttal policies in the email (\[DB Track NeurIPS 2025] Rebuttal format clarification) stating: "We prohibit use of any links in the rebuttal (including but not limited to anonymous or non-anonymous URL links), or updating your submitted data or code repositories," we are not allowed to provide additional files beyond what was initially uploaded during the review process.
> >
> > For your reference, here is a sample output from GPT4.1 on the regional change captioning task:
> >
> > ```json
> > {"id": "regional_caption_85", "response": "**2012-2014**  \nAt the start of this period, the red-demarcated zone is characterized by largely barren, unvegetated land, showing light earth tones with minimal surface disturbance and no visible built constructions. Throughout this interval, there are no apparent changes: vegetation coverage, land condition, and absence of infrastructure or utilities remain consistent.\n\n**2014-2016**  \nAt the beginning of this block, the area continues to reflect the same undeveloped state—open ground with negligible vegetation and no constructed features. No new structures, clearings, or infrastructural elements emerge within this span; the landscape exhibits ongoing non-change.\n\n**2016-2020**  \nThe interval opens with the terrain still appearing vacant and largely unaltered, mirroring previous years with minor, patchy vegetation in an otherwise open lot. Marked transformation occurs during this period: by 2020, extensive construction activity is evident. Multiple large building footprints and associated paved surfaces now occupy much of the formerly open area, accompanied by organized parking lots and clear street access points leading into the interior of the zone. All former characteristics of barren land are replaced by developed infrastructure, signaling a complete transition from vacant to built-up land cover.", "request": {"id": "regional_caption_85", "image_list": ["0_Bend/m_4412162_ne_10_res_7_1/m_4412162_ne_10_1_20120610_res_7_1.png", "0_Bend/m_4412162_ne_10_res_7_1/m_4412162_ne_10_1_20140605_res_7_1.png", "0_Bend/m_4412162_ne_10_res_7_1/m_4412162_ne_10_1_20160621_res_7_1.png", "0_Bend/m_4412162_ne_10_res_7_1/m_4412162_ne_10_060_20200625_res_7_1.png"], "time_stamps": ["20120610", "20140605", "20160621", "20200625"], "ground_truth": "2012-2016, the scene shows a cleared, mostly bare lot with patchy ground cover and a minor access road at the bottom; no permanent buildings are present.  \n2016-2020, a large light-toned roofed structure is constructed across the top-center, with paved parking and new internal roadways filling the remainder of the parcel, converting the previously vacant land into a fully developed complex.", "task": "Regional Caption", "cls_description": ""}}
> > ```
> >
> > We confirm that we will release all models' outputs upon the public release of our benchmark.

---

> > ### Author Response · Authors · 2025-08-06
> >
> > > Q2: Ablation studies.
> >
> > Thank you for your suggestions. We conducted extensive ablations across three key dimensions to better understand our framework's contributions, with results shown as follows:
> >
> > |Setting|OA-AVG|BCA-Report|CSE-Report|RCC|DTC|IoU|
> > |---|---|---|---|---|---|---|
> > |Qwen2.5-VL (baseline)|23.3|2.94|1.73|3.21|2.85|-|
> > |**Training Paradigm**||||||
> > |+ Joint Training|11.6|0.74|0.65|0.96|1.10|27.45|
> > |+ Routing-based Separate Training (Ours)|33.3|3.47|2.51|3.98|3.40|29.06|
> > |**Hyperparams**||||||
> > |LoRA rank - 32|32.7|3.49|2.43|3.92|3.34|28.63|
> > |LoRA rank - 128|33.1|3.44|2.56|3.94|3.37|28.76|
> > |LoRA rank - 64 (Ours)|33.3|3.47|2.51|3.98|3.40|29.06|
> > |**Data**||||||
> > |Data 20%|26.7|3.11|2.33|3.52|3.19|24.52|
> > |Data 50%|28.4|3.21|2.37|3.66|3.21|27.03|
> > |Data without Regional & Dense|32.5|3.42|2.44|3.32|2.95|29.06|
> > |Data without Regional|32.7|3.44|2.48|3.64|3.37|29.06|
> > |Data without Dense|33.1|3.41|2.52|3.95|3.18|29.06|
> > |Full Data (Ours)|33.3|3.47|2.51|3.98|3.40|29.06|
> >
> > **Training Paradigm:** Our routing-based separate training strategy significantly outperforms joint training (33.3% vs 11.6% overall accuracy), confirming the hypothesis about optimization conflicts between VQA and segmentation tasks \[1]. The joint training approach suffers severe performance degradation across all metrics, particularly in report generation tasks (BCA-Report: 0.74 vs 3.47, CSE-Report: 0.65 vs 2.51), validating our training paradigm design choice.
> >
> > **Hyperparameter:** LoRA rank ablation demonstrates that rank-64 achieves optimal efficiency-accuracy trade-off (33.3% overall accuracy), consistent with general domain practices. The performance remains stable across ranks 32-128, justifying our settings.
> >
> > **Data:** Progressive data reduction shows substantial performance drops (26.7% with 20% data vs 33.3% with full data), emphasizing the importance of our comprehensive training data. Task-specific data ablation reveals targeted impacts: removing regional data primarily affects regional change caption performance (RCC: 3.64 vs 3.98), while removing dense temporal captioning data mainly impacts dense temporal captioning (DTC: 3.18 vs 3.40). When both are removed simultaneously, both caption tasks suffer notable degradation (RCC: 3.32, DTC: 2.95), demonstrating that each task-specific training data contributes most effectively to its corresponding capability while providing complementary benefits to the overall framework.
> >
> > These ablations confirm that our primary contribution lies in the carefully curated dataset that enables existing multimodal large language models to adapt to long-term urban analysis through direct instruction tuning, following the data-centric paradigm established by existing high-impact benchmark papers in remote sensing \[2-5]. We will incorporate your valuable suggestions into the revised version.
> >
> > \[1] Zhang, Z., et al. (2024). Psalm: Pixelwise segmentation with large multi-modal model. *ECCV 2024.*
> >
> > \[2] Kuckreja, K., et al. (2024). Geochat: Grounded large vision-language model for remote sensing. *CVPR 2024.*
> >
> > \[3] Li, X., Ding, J., & Elhoseiny, M. (2024). Vrsbench: A versatile vision-language benchmark dataset for remote sensing image understanding. *NeurIPS 2024 D\&B Track.*
> >
> > \[4]Muhtar, D., et al. (2024, September). Lhrs-bot: Empowering remote sensing with vgi-enhanced large multimodal language model. *ECCV 2024.*
> >
> > \[5] Irvin, J. A., Liu, et al. (2024). Teochat: A large vision-language assistant for temporal earth observation data. *ICLR 2025.*
> >
> > > Q3: DB Track.
> >
> > Thank you for your perspective. We fully agree that benchmark contributions should maintain high standards and provide significant value to the research community. Our work addresses a critical gap in multimodal large language models for long-term remote sensing understanding by introducing the first comprehensive benchmark for long-term temporal understanding in remote sensing, moving beyond existing bi-temporal limitations. The low performance of state-of-the-art models (34.1% accuracy for the best commercial model) demonstrates both the necessity and challenge level of our benchmark, while our systematic evaluation across 17 models provides important insights for the community. We believe this dataset- and benchmark-centric contribution, with its rigorous annotation pipeline and comprehensive task taxonomy, establishes a valuable foundation for advancing multimodal temporal understanding research in the remote sensing community.

---

> > ### Author Response · Authors · 2025-08-06
> >
> > > &#x20;Q3: Results on other benchmarks
> >
> > Thank you for this valuable suggestion. We evaluated DVLChat across multiple established remote sensing benchmarks to assess its generalization capabilities:
> >
> > |Method|VRS-Bench(VQA, Acc %)|VRS-Bench(Captioning, CLAIR score)|GeoChat(UC Merced, Acc %)|GeoChat(AID, Acc %)|TEOChat(RQA, Acc %)|TEOChat(RTQA, Acc %)|
> > |---|---|---|---|---|---|---|
> > |Qwen2.5-VL 3B|38.6|0.5526|56.6|55.2|88.6/87.5|55.7|
> > |DVLChat-3B|39.3|0.5633|59.2|57.1|91.3/89.7|62.1|
> > |Qwen2.5-VL 7B|43.5|0.6374|72.3|64.6|76.7/87.3|59.9|
> > |DVLChat-7B|44.7|0.6790|74.4|66.0|91.5/90.0|64.2|
> >
> > We evaluated across diverse tasks, including VRS-Bench \[1]'s VQA (accuracy) and captioning (CLAIR score) tasks, GeoChat \[2]'s classification tasks on UC Merced and AID datasets (accuracy), and TEOChat \[3]'s RQA tasks on xBD/S2Looking subsets and RTQA tasks on QFabric (all accuracy-based).
> >
> > DVLChat demonstrates consistent improvements across all task types and evaluation metrics, with clear gains in VQA (7B +1.2%), captioning (7B CLAIR score +0.04), classification (7B +2.1% on UC Merced, +1.4% on AID), and out-of-distribution multi-temporal tasks (7B +4.3% for RTQA of TEOChat), demonstrating robust cross-domain transferability rather than task-specific overfitting. We attribute this superior performance to our comprehensive data curation strategy that encompasses both single-image and multi-temporal image understanding across diverse urban scenarios, spanning 42 major U.S. cities with broad geographic coverage. This extensive spatial and temporal diversity, combined with LLM-enhanced linguistic diversity while maintaining terminological accuracy through expert validation, produces training data that naturally enhances model generalization capabilities. The resulting DVLChat model demonstrates strong transferability across diverse remote sensing applications, validating that our geographically comprehensive and task-diverse data-centric approach successfully builds models with enhanced cross-domain capabilities.
> >
> > We will also incorporate the detailed analysis of transferability in the revised version. We thank the reviewer for raising such valuable suggestions.
> >
> > \[1] Li, X., et al. (2024). Vrsbench: A versatile vision-language benchmark dataset for remote sensing image understanding. *NeurIPS 2024 D\&B Track.*
> >
> > \[2] Kuckreja, K., et al. (2024). Geochat: Grounded large vision-language model for remote sensing. *CVPR 2024.*
> >
> > \[3] Irvin, J. A., Liu, E. R., et al. (2024). Teochat: A large vision-language assistant for temporal earth observation data. *ICLR 2025.*
> >
> > > Q4: Automated code implementation.
> >
> > Thank you for your suggestions regarding the evaluation code and testing process. To clarify, our evaluation metrics primarily consist of standard accuracy calculations for multiple-choice questions \[1], LLM-as-judge for open-ended generation tasks following established practices in the vision-language community \[2]\[3], and conventional cIoU and F1-score metrics for referring segmentation \[4]\[5]. These metrics are built upon well-established implementations from the general domain, with many components being reimplementations of existing evaluation frameworks, ensuring no inherent reproducibility challenges.
> >
> > Our initial evaluation process employed separate prediction and metric computation steps. According to your suggestions, we have developed comprehensive automated bash scripts that integrate model inference and evaluation metric computation into a single end-to-end pipeline, providing complete automation of the testing process. However, due to NeurIPS's rebuttal policy stating "We prohibit use of any links in the rebuttal (including but not limited to anonymous or non-anonymous URL links), or updating your submitted data or code repositories," we are not allowed to provide direct access during this phase. We commit to releasing the complete automated codebase with detailed documentation upon the benchmark's public release, facilitating easier reproduction of all benchmark results.
> >
> > \[1] Wang, F., et al. Xlrs-bench: Could your multimodal llms understand extremely large ultra-high-resolution remote sensing imagery?. *CVPR 2025.*
> >
> > \[2] Yuan, Y., et al. (2025). Videorefer suite: Advancing spatial-temporal object understanding with video llm. *CVPR 2025.*
> >
> > \[3] Cai, M., et al. (2024). Vip-llava: Making large multimodal models understand arbitrary visual prompts. *CVPR 2024.*
> >
> > \[4] Lai, X., et al. (2024). Lisa: Reasoning segmentation via large language model. *CVPR 2024.*
> >
> > \[5] Chen, H., et al. (2024). ChangeMamba: Remote sensing change detection with spatiotemporal state space model. *TGRS 2024.*
> >
> > &nbsp;
> > &nbsp;
> >
> >
> > We hope these further clarifications address the concerns you have raised. We will follow all of your suggestions to further improve the quality of our paper. Once again, we sincerely thank the reviewer for your time, careful consideration, and valuable suggestions!

---

### Note · Authors · 2025-08-12

We would like to express our profound thanks to all reviewers for the valuable insights they have provided on our work. In addition, we sincerely appreciate the constructive comments and suggestions from reviewers. We are particularly grateful that their reviews have highlighted the following shared recognitions:

* (All reviewers) **DVL-Suite fills the critical gap in the remote sensing field** by providing the first comprehensive framework for long-term temporal understanding

* (All reviewers) **Extensive experimental evaluation across diverse model families** that reveals significant challenges in current MLLMs

* (All reviewers) **Large-scale, well-annotated dataset**

* (R#8uLB, R#xvx3, R#vVPe) **Well-designed practical tasks** that systematically evaluate models' urban understanding capabilities

* (R#qr7v, R#xvx3, R#vVPe) Clear paper organization and presentation

* (R#8uLB, R#vVPe) DVLChat baseline model development advances long-term temporal understanding



While we are encouraged by the positive feedback, we also deeply value the constructive concerns raised by reviewers to strengthen our contribution. In our point-by-point responses, we have provided detailed clarifications of our core contributions and methodological choices. We commit to incorporating all promised improvements from our rebuttal into the revised manuscript to enhance the work's impact on both remote sensing and broader machine learning communities. Specifically:

- We will include the NAIP data selection justifications provided in our rebuttal to better position our work.

- We will supplement the paper with additional experimental results from our rebuttal, including different-sized MLLMs as DVLChat base models, EarthDial results, data-oriented ablation studies, and transferability evaluations on other benchmarks.

- All other suggested improvements promised in our rebuttal responses will be fully incorporated into the revised version.



Finally, we express our sincere gratitude to all reviewers for their time and effort in the review process. We are particularly pleased that reviewers unanimously recognize the important value and impact of our core dataset and benchmark contributions. We believe this work merits publication and can genuinely drive progress in this important subfield from a practical dataset and benchmark perspective.



Sincerely,

Authors of Submission #300

---

### Decision · Program_Chairs · 2025-09-18

**Decision:**

Accept (poster)

**Comment:**

AC and reviewers thank the authors for the rebuttal. After the rebuttal, reviewers converged toward acceptance (one strong accept, two accept, one borderline accept). Overall the paper presents DVL-Suite, a well-scoped benchmark for dynamic city understanding with broad tasks and thorough evaluations. Reviewers found the contribution timely and useful; Most of initial concerns on baseline coverage, scaling effects, and data-selection/seasonality/metric clarity were reasonably addressed in rebuttal via added baselines and scaling results, clarifications on NAIP selection. There are still several small concerns raised by reviewer qr7v, and the authors are encouraged to further address these points and incorporate the reviewer-requested additions and clarifications (including those points from other reviewers) in the camera-ready. Considering all these factors, the AC would like to accept the paper. Congrats!